Biogeosciences Discussions Open Access EGU

# Isotopic fractionation of carbon during uptake by phytoplankton across the South Atlantic subtropical convergence

Robyn E. Tuerena[1], Raja S. Ganeshram[1], Matthew P. Humphreys[2†], Thomas J. Browning[3‡], Heather Bouman[3], and Alexander. P. Piotrowski[4],

[1]School of GeoSciences, University of Edinburgh, Edinburgh, UK
[2]Department of Earth Sciences, University of Oxford, Oxford, UK
[3]Ocean and Earth Science, University of Southampton, Southampton, UK
[4]School of Geosciences, University of Cambridge, Cambridge, UK

[†] Now at: School of Environmental Sciences, University of East Anglia, Norwich, UK
[‡] Now at: GEOMAR Helmholtz Centre for Ocean Research, Kiel, Germany

*Correspondence to*: Robyn E. Tuerena (r.tuerena@ed.ac.uk)

**Abstract.** The stable isotopic composition of particulate organic carbon ($\delta^{13}C_{POC}$) in the surface waters of the global ocean can vary with the aqueous $CO_2$ concentration ($[CO_{2(aq)}]$) and affects the trophic transfer of carbon isotopes in the marine food web. Other factors such as cell size, growth rate and carbon concentrating mechanisms decouple this observed correlation. Here, the variability in $\delta^{13}C_{POC}$ is investigated in surface waters across the south subtropical convergence (SSTC) in the Atlantic Ocean, to determine carbon isotope fractionation ($\varepsilon_p$) by phytoplankton and the contrasting mechanisms of carbon uptake in the subantarctic and subtropical water masses. Our results indicate that cell size is the primary determinant of $\delta^{13}C_{POC}$ across the Atlantic SSTC in summer. Combining cell size estimates with $CO_2$ concentrations, we can accurately estimate $\varepsilon_p$ within the varying surface water masses in this region. We further utilize these results to investigate future changes in $\varepsilon_p$ with increased anthropogenic carbon availability. Our results suggest that smaller cells, which are prevalent in the subtropical ocean, will respond less to increased $[CO_{2(aq)}]$ than the larger cells found south of the SSTC and in the wider Southern Ocean. In the subantarctic water masses, isotopic fractionation during carbon uptake will likely increase, both with increasing $CO_2$ availability to the cell, but also if increased stratification leads to decreases in average community cell size. Coupled with decreasing $\delta^{13}C$ of $[CO_{2(aq)}]$ due to anthropogenic $CO_2$ emissions, this change in isotopic fractionation and lowering of $\delta^{13}C_{POC}$ may propagate through the marine food web, with implications for the use of $\delta^{13}C_{POC}$ as a tracer of dietary sources in the marine environment.

## 1 Introduction

The marine environment is undergoing rapid changes as atmospheric carbon dioxide increases, with the greatest change occurring in the upper ocean (Gruber et al., 1999; Sabine and Tanhua, 2010). Anthropogenic carbon inputs to the atmosphere are causing ocean warming (Cheng et al., 2019), changes to upper ocean stratification (Bopp et al., 2001; Capotondi et al., 2012) and altered distributions of nutrients and carbon (Khatiwala et al., 2013; Quay et al., 2003; Gruber et al., 2019). Marine phytoplankton are diverse, and are already responding to ocean warming, including changes to productivity (Behrenfeld et al., 2006, Arrigo and van Dijken, 2015), the length of growing season (Henson et al., 2018) and phytoplankton cell size (Finkel et al., 2010). Alterations to phytoplankton diversity and/or productivity will likely have knock-on effects on marine food web dynamics.

During photosynthesis, marine phytoplankton take up aqueous $CO_2$ ($CO_{2(aq)}$) and convert it into organic carbon. In this process the lighter isotope ($^{12}C$) is preferentially consumed, leaving the residual aqueous pool increasingly enriched in the heavier isotope. The stable carbon isotopic composition of marine phytoplankton is determined by the uptake fractionation ($\varepsilon_p$), which is influenced by ambient environmental conditions and phytoplankton cell physiology. Therefore the $\delta^{13}C$ of marine plankton can indicate the controlling mechanisms behind carbon uptake which led to the use of $\delta^{13}C_{POC}$ as a potential proxy to reconstruct surface water $CO_{2(aq)}$ concentration ($[CO_{2(aq)}]$) of past climates (Freeman and Hayes, 1992, Jasper et al., 1994).

The $\delta^{13}C$ of particulate organic carbon ($\delta^{13}C_{POC}$) varies over relatively large oceanic areas and has been found to inversely correlate with $[CO_{2(aq)}]$ (the principal carbon source) in surface waters (Rau et al., 1991; Sackett et al., 1965). High $[CO_{2(aq)}]$ can lead to greater discrimination against $^{13}C$ as the light isotope is preferentially consumed by phytoplankton. The low temperature waters of the Southern Ocean and their high $[CO_{2(aq)}]$ lead to negative $\delta^{13}C$ excursions in marine plankton there (Sackett et al., 1964). Although this relationship holds true to first order over global datasets (Rau et al., 1989), in many marine environments the local variability in $\delta^{13}C_{POC}$ can be attributed to other mechanisms.

Phytoplankton growth rate, cell size and cell geometry are also important controls on $\delta^{13}C_{POC}$ in surface waters (Bidigare et al., 1997; Francois et al., 1993; Popp et al., 1998; Laws et al., 1995; Villinski et al., 2000). These ecophysiological factors decouple the observed relationship between $\delta^{13}C_{POC}$ and $[CO_{2(aq)}]$, limiting the reliability of $\delta^{13}C_{POC}$ as a palaeoproxy. This is particularly true in areas where $[CO_{2(aq)}]$ is lower or less variable, as other factors have been found to be more important for determining the degree of isotopic fractionation (Henley et al., 2012; Lourey et al., 2004; Popp et al., 1998).

The carbon fixation pathway can vary amongst phytoplankton species through the assimilation of bicarbonate via active transport as opposed to diffusive $CO_2$ uptake. In general, more negative excursions in $\delta^{13}C_{POC}$ are associated with diffusive entry of $CO_2$, whereas CCMs or diffusive limitation of carbon supply lead to more positive $\delta^{13}C_{POC}$ (Raven et al., 2008). When $[CO_{2(aq)}]$ falls below a critical level, the active transport of $CO_2$ into the cell can enrich $\delta^{13}C_{POC}$, but this has been found to be proportional to carbon demand or growth rate (Popp et al., 1998). CCMs occur in most cyanobacteria, increasing $CO_2$ at the site of Rubisco activity (Raven et al., 2008).

All of the processes which alter the uptake fractionation of [$CO_{2(aq)}$] will ultimately determine the isotopic variability in carbon at the base of the marine food web. Thus, the factors that are sensitive to ongoing climate change need to be better understood in order to accurately use $\delta^{13}C_{POC}$ as a tracer of dietary sources and in food web studies.

In this study we investigate the mechanisms for isotopic fractionation in $\delta^{13}C_{POC}$ resulting from carbon uptake and biological production in the upper ocean. We report data from a full transect across the south subtropical convergence (SSTC) in the Atlantic basin, which captures a region of productive open ocean. The cruise sampled both subantarctic and subtropical regimes with contrasting limiting nutrient environments and community structure (Browning et al., 2014). The parameters [$CO_{2(aq)}$] and $\delta^{13}C_{DIC}$, together with chlorophyll-a and other diagnostic phytoplankton pigments are used collectively to disentangle the processes which fractionate $\delta^{13}C_{POC}$ as a response to algal uptake of [$CO_{2(aq)}$] across this region. We find the community cell size, as estimated using phytoplankton pigment composition, to be the primary determinant of $\delta^{13}C_{POC}$ across the SSTC, with smaller cell sizes increasing the carbon availability for fixation. The results from the field study are used to understand/infer how $\delta^{13}C_{POC}$ in this region may change into the future with ongoing climate change.

## 2. Methods

### 2.1 Carbon concentrations and isotopic measurements

Samples were collected onboard the RRS James Cook between December 2011 and February 2012 (JC068), as part of the GEOTRACES A10 transect of the South Atlantic. An east to west transect was conducted with upper ocean sampling at each station. Standard CTD measurements and water sampling were performed using a stainless steel rosette equipped with a full sensor array and 24 × 20-litre OTE bottles. Salinity, temperature and depth were measured using a CTD system (Seabird 911+) and salinity was calibrated on-board with discrete samples using an Autosal 8400B salinometer (Guildline).

Measurements of total $CO_2$ ($TCO_2$) and total alkalinity (TA) were carried out at sea within 24 hours of collection. Samples were warmed in a water bath at 25 °C for an hour before analysis. A set volume of the sample is acidified by addition of excess 10% phosphoric acid, which converts all inorganic C species to $CO_2$. This is carried into the coulometric cell by an inert carrier gas ($CO_2$-free $N_2$ that is first passed through a magnesium perchlorate and Ascarite II scrubber), and a coulometric titration determines the amount of $CO_2$, which is equal to $TCO_2$. Small increments of 0.1 M hydrochloric acid are added to a separate subsample and the amount added to reach the carbonic acid equivalence point is equal to the TA (Humphreys, 2015). Regular measurements of both $TCO_2$ and TA were made from batch 114 Certified Reference Material (CRM) from A. G. Dickson (Scripps Institution of Oceanography; Dickson et al., 2003) and used to calibrate the results. To obtain the final results in units of μmol $kg^{-1}$, a correction for density (ρ) due to salinity variation was then applied using salinity measured from Niskin bottle samples (Zeebe et al., 2001). Duplicate samples were taken from the same Niskin bottle and analysed consecutively.

[$CO_{2(aq)}$] was calculated from measured TA and DIC using $CO_2$SYS v1.1 (Lewis and Wallace, 1998; van Heuven et al., 2011). Equilibrium constants were evaluated following Mehrbach et al. (1973) for carbonic acid and Dickson (1990) for bisulfate, and we used the boron:chlorinity ratio of Lee et al. (2010).

Samples for the measurement of the stable isotopes of carbon in dissolved inorganic carbon ($\delta^{13}C_{DIC}$) were collected from the stainless steel rosette. Samples were taken into 250 mL glass bottles with ground glass stoppers. Water was drained directly into the sample bottle using silicone tubing to the bottom of the bottle to eliminate bubble formation. The bottle and cap were rinsed once with water from the rosette bottle before overflowing the sample bottle by at least 1 bottle volume before withdrawing the silicone tube, carefully avoiding bubble formation. The stopper was then placed in the bottle and then removed so that 2.5 mL of sample could be removed to allow for thermal expansion, and 50 mL of 100% $HgCl_2$ added to halt any biological activity. The stoppers and the inside of the neck of the bottles were dried before the stopper, coated with vacuum grease, was replaced and secured with a foam insert and plastic cover. The samples were then shaken to disperse the $HgCl_2$ and stored at 4 °C until analysis. Samples were measured using a Thermo MAT253 stable isotope mass spectrometer at the University of Cambridge. $\delta^{13}C_{CO2}$ was determined from $\delta^{13}C_{DIC}$ and absolute temperature ($T_k$), using $\delta^{13}C_{CO2} = \delta^{13}C_{DIC} + 23.644 - 9701.5/T_k$ (Rau et al., 1996).

Particulate samples were collected onto ashed, pre-weighed GF/F microfibre filters (0.7 µm pore size, 25 mm diameter). Two to four litres of water were collected from the biological rosette in the surface 400 m depending on chlorophyll levels detected by the CTD fluorometer. The samples were pressure filtered simultaneously using an 8-way manifold system. Once the total volume for each depth was filtered, the filters were extracted from the filter holder, placed in labelled aluminium foil and dried at 50 °C for ~12 hours. Once dried, filters were folded and stored in plastic sample bags at -20 °C. To remove carbonates prior to analysis, filters were wetted with Milli-Q water, fumed with 70% HCl for 48 hours in a desiccator, dried at 50 ºC and then folded into tin capsules. The filters were analysed using a Carlo Erba NA 2500 elemental analyser in-line with a VG PRISM III isotope ratio mass spectrometer for elemental POC/PN and $\delta^{13}C_{POC}$ and $\delta^{15}N_{PN}$. All $\delta^{13}C_{POC}$ data presented in this study are in the delta per mil notation versus V-PDB (‰ $_{VPDB}$).

**2.2 Cell size calculations**

Phytoplankton pigments were analysed by High Pressure Liquid Chromatography (HPLC) analysis. Between 500 and 2000 ml of seawater was filtered through 25 mm GF/F filters. The filters were placed in 2 ml cryovials and flash frozen in liquid nitrogen. Filters were then transferred to a -80 °C freezer for longer term storage. Pigment extracts were analysed using a reverse-phase HPLC column using a Thermo Finnigan HPLC instrument at the National Oceanography Centre Southampton (Gibb et al., 2000). Phytoplankton pigments were extracted in 3 to 5 m1 90% acetone by ultrasonication and centrifugation. Extracts were loaded into a chilled autosampler prior to injection into the HPLC system. Pigments were detected by absorbance at 440 nm, and identified by diagnostic retention times. The resulting pigment assemblage was used to estimate the fractional contribution of the three size classes (micro-, nano- and picophytoplankton) to total chlorophyll-a pigment concentration (Bricaud et al. 2004; Uitz et al. 2008).

## 3. Results

### 3.1 Oceanographic setting

The SSTC is characterised by the convergence of contrasting biogeochemical regimes. In the colder Subantarctic Surface Waters (SASW), located south of the SSTC, concentrations of macronutrients are elevated and primary production is primarily limited by iron availability (Browning et al., 2014). The subtropical waters to the north of the SSTC are associated with the South Atlantic subtropical gyre and are principally macronutrient limited, or possibly macronutrient-iron co-limited (Browning et al., 2014, 2017). The three subtropical water masses (Agulhas Current (AC), South Atlantic Central Water (SACW) and Brazil Current (BC)) can be readily identified with warmer temperatures and higher salinities (Figure 1).

Higher [$CO_{2(aq)}$] is associated with the lower temperatures of the SASW. $\delta^{13}C_{CO2}$ inversely correlates with [$CO_{2(aq)}$], and low $\delta^{13}C_{CO2}$ is associated with higher [$CO_{2(aq)}$] and lower temperatures (Figure 1). $\delta^{13}C_{CO2}$ is highest on the western boundary in the BC and in the Rio Plata outflow. $\delta^{13}C_{POC}$ across 40°S ranges from -25 to -20‰ indicating a predominantly marine source (e.g. Rau et al., 1989).

Satellite images of surface chlorophyll concentrations across this region indicate elevated standing stocks of phytoplankton in comparison to the South Atlantic gyre and subantarctic waters further south (Browning et al., 2014). Chlorophyll concentrations peak between austral spring and summer, and the south subtropical convergence (SSTC) moves south as a result of the expansion of the Agulhas and Brazil Currents. Depth profiles showed that the subantarctic waters have elevated and uniform chlorophyll concentrations (0.2-0.9 mg m$^{-3}$). Conversely in the subtropical waters a deep chlorophyll maximum is formed, with low surface concentrations of chlorophyll (<0.2 mg m$^{-3}$) and macronutrients (Tuerena et al., 2015).

### 3.2 $\delta^{13}C_{POC}$ variability

If $\delta^{13}C_{POC}$ is determined principally by changing ambient [$CO_{2(aq)}$] and not influenced by cell physiology, such as growth rates and cell size the $\delta^{13}C_{POC}$ can often be predicted by sea surface temperature variability (Rau et al., 1989). In this study, $\delta^{13}C_{POC}$ is compared to a model from Rau et al. (1996), which predicts the carbon isotope fractionation ($\varepsilon_p$) and $\delta^{13}C_{POC}$ where photosynthesis is strictly based on the passive diffusion of $CO_2$ into marine phytoplankton cells.

Modelled $\delta^{13}C_{POC}$ was calculated using the diffusion model of Rau et al. (1996), where

$$\delta^{13}C_{POC} = \delta^{13}C_{CO2} - \varepsilon_f + (\varepsilon_f - \varepsilon_d)\,\frac{Q_s}{[CO_{2(aq)}]}\left(\frac{r}{D_T\left(1+\frac{r}{r_k}\right)} + \frac{1}{P}\right), \quad (1)$$

where $\varepsilon_f$ = intracellular enzymatic isotope fractionation (‰), $\varepsilon_d$ = Diffusive isotope fractionation of $CO_{2(aq)}$ in seawater (‰), $Q_s$ = $CO_2$ uptake rate per unit cell surface area (mol-C m$^{-2}$ s$^{-1}$), [$CO_{2(aq)}$]= ambient $CO_{2(aq)}$ concentration (mol m$^{-3}$), r = cell radius (m), $D_T$ = temperature dependent diffusion rate of $CO_2$ (m$^2$ s$^{-1}$), $r_k$ = reacto-diffusive length (m) and P = cell wall permeability to $CO_2$ (m s$^{-1}$). $Q_s$ was determined from

$$Q_s = \frac{(\gamma_c\,\mu_i)}{4\pi r^2} \quad (2)$$

where $\gamma_c$ = carbon content per cell (mol-C), $\mu_i$ = cell growth rate ($s^{-1}$).

Using this model we tested how $\delta^{13}C_{POC}$ and $\varepsilon_p$ vary as a function of [$CO_{2(aq)}$], temperature, growth rate and cell size. For each of these parameters we used the base values of Rau et al. (1996) unless specified.

$\varepsilon_p$ for measured and modelled $\delta^{13}C$ was calculated as:

$$\varepsilon_p = \delta^{13}C_{CO2} - \delta^{13}C_{POC} \quad (3)$$

To understand under what conditions the ambient [$CO_{2(aq)}$] plays a dominant role in the determination of $\delta^{13}C_{POC}$ in surface waters across the frontal region, the relationships between $\delta^{13}C_{POC}$, $\delta^{13}C_{CO2}$, and $CO_{2(aq)}$ were compared to modelled estimates for passive diffusion (Figure 2). SASW samples fall close to the modelled estimates and subtropical samples decouple from the modelled trend to a much higher degree. In Figure 2a, [$CO_{2(aq)}$] and $\delta^{13}C_{CO2}$ in the SASW have a significant negative correlation ($r=-0.77$, $n=12$, $p=0.003$), and correspond to the model trends, with lower concentrations resulting in higher $\delta^{13}C_{DIC}$ and $\delta^{13}C_{CO2}$. There are no significant correlations between $\delta^{13}C_{POC}$ and [$CO_{2(aq)}$] or $\delta^{13}C_{CO2}$ (Figure 2, $p>0.05$). These data suggest that although [$CO_{2(aq)}$] may play a part in determining the $\delta^{13}C_{POC}$ in the SASW, other factors cause deviation away from a significant correlation, with the relationship increasingly decoupled in subtropical waters.

To investigate the spatial variability across the SSTC, [$CO_{2(aq)}$], and $\delta^{13}C_{CO2}$ were plotted against longitude and compared to model estimates based on temperature alone and using a constant cell size of 10 µm (Figure 3a and b). Temperature can predict the spatial variability in [$CO_{2(aq)}$], and $\delta^{13}C_{CO2}$, however ambient [$CO_{2(aq)}$] is lower and $\delta^{13}C_{CO2}$ higher than model estimates, resulting from biological production and the isotopic disequilibrium between the ocean and atmosphere (Gruber et al., 1997). $\delta^{13}C_{POC}$ is predicted using ambient temperature and also measurements of [$CO_{2(aq)}$] and $\delta^{13}C_{CO2}$ (Figure 3c). There is no correlation between measured and modelled $\delta^{13}C_{POC}$ across the transect, suggesting there are other controlling factors which determine the $\delta^{13}C_{POC}$ variability.

To test whether cell size (and thus cellular surface area to volume ratio) plays an important role in determining $\delta^{13}C_{POC}$, we estimated the change in phytoplankton size classes across the transect. Using phytoplankton pigment data we calculated the relative proportion of pico-, nano- and microphytoplankton size fractions to total chlorophyll-a biomass (Bricaud et al. 2004; Uitz et al., 2008). Picophytoplankton were dominant in the subtropical environments, contributing between 30-40% of the pigment biomass at the core of these water masses (Figure 4). In contrast, nano- and microphytoplankton were more dominant in the SASW and close to the Rio Plata outflow.

An estimate of the approximate average community cell size was calculated by defining a specific cell size for each of the three defined size classes (picophytoplankton=1 µm, nanophytoplankton=5 µm, microphytoplankton= 50 µm, Bricaud et al. 2004). The central size values for each class were divided by two to approximate the average community cell radius (picophytoplankton=0.5 µm, nanophytoplankton=2.5 µm, microphytoplankton= 25 µm). This method provides only a rough indicator of the community cell radius, as size class is being represented by one unique size for each algal group, however this enables one single parameter to be used to characterise the size structure of the algal population, which is important for the purposes of this study.

Estimated average cell radii were smaller in the subtropical water masses compared to the SASW (Figure 5). Increasing the average cell size (and thus decreasing SA:V) has the potential to reduce carbon isotope fractionation during uptake by passive diffusion and thus increase $\delta^{13}C_{POC}$, by reducing the ability of the cell to discriminate between the two isotopes. This has been found in modelled, experimental and environmental studies (Popp et al., 1998, Pancost et al., 1997, Rau et al., 1990, 1996). When $\delta^{13}C_{POC}$ is modelled using temperature, SASW measurements fall between model estimates for a cell radius of 10-15 µm (Figure 6a). Conversely, the subtropical samples have higher proportions of picoplankton (<2 µm), and decrease to lower $\delta^{13}C_{POC}$ than those predicted using temperature alone, demonstrating that cell size is likely a controlling factor in $\delta^{13}C_{POC}$ determination. The average community cell radius in open ocean samples were compared to $\delta^{13}C_{POC}$ (Figure 6b), and a significant positive correlation was observed size ($r=0.74$, $n=30$, $p<0.001$).

The samples with a larger estimated community cell size on both the east and western margins were not included in this correlation analysis as they show a significant offset from this relationship (Figure 6b). These samples have a larger estimated cell size compared to measured $\delta^{13}C_{POC}$ and suggest there is a possible terrestrial influence, either with the supply of allochthonous material, the presence of grazers and/or significant shifts in species assemblage to a higher abundance of micro plankton (Browning et al., 2014). The significant positive correlation between cell size and $\delta^{13}C_{POC}$ for open ocean waters suggests that cell size is the primary factor influencing $\delta^{13}C_{POC}$ in the surface waters across the SSTC. We further test the relationship between $\delta^{13}C_{POC}$ and cell size by predicting changes in $\delta^{13}C_{POC}$ using temperature and cell size measurements (black crosses in Figure 6b). We find good agreement between modelled and measured data points, demonstrating the importance of cell size in estimating $\delta^{13}C_{POC}$.

South of the subtropical front, the phytoplankton community is dominated by haptophytes (Browning et al., 2014). A lower species diversity south of the front may explain the closer alignment between $[CO_{2(aq)}]$ and $\delta^{13}C_{POC}$, as other factors are less significant in influencing $\varepsilon_p$. Recent work has highlighted the interspecies differences in carbon uptake fractionation and their influence on bulk $\delta^{13}C_{POC}$ (Hansman and Sessions, 2016). Our results suggest a changing community cell size deviates $\delta^{13}C_{POC}$ from expected trends with $[CO_{2(aq)}]$. In this open ocean environment, using estimates of cell size in addition to $[CO_{2(aq)}]$, we can predict variability in $\delta^{13}C_{POC}$.

## 4. Discussion

### 4.1 Carbon uptake fractionation across the 40°S transect

The biological fractionation of carbon isotopes during uptake by phytoplankton can be estimated using $\varepsilon_p \sim \delta^{13}C_{CO2} - \delta^{13}C_{POC}$ (Freeman and Hayes, 1992). This fractionation comprises both the $CO_2$ fixation during photosynthesis, which utilizes the enzyme Rubisco (~-22 to -31‰), and is also determined by the factors which limit the external supply of $CO_2$ to the enzyme. Therefore, the more $CO_2$-limited the cell, the less the isotopic fractionation of $CO_2$ fixation will be expressed. These limiting factors include ambient $[CO_{2(aq)}]$ (Baird et al., 2001), growth rates (Laws et al., 1995, Popp et al 1998), cell size or geometry

(Popp et al., 1998), light availability and day length (Laws et al., 1995, Burkhardt et al., 1995), utilization of $HCO_3^-$ in replacement of $CO_2$ (Sharkey and Berry, 1985) and species variability (Falkowski, 1991).

Empirical estimates of $\varepsilon_p$ range between 10-18‰, with the highest fractionation observed in the Southern Ocean where $[CO_{2(aq)}]$ is highest, increasing to over 20 µM in surface waters (Young et al., 2013). Over the Atlantic SSTC we measure an $\varepsilon_p$ range of 12-17‰. In the subtropical water masses north of the SSTC, the average $\varepsilon_p$ is 1‰ higher than in the SASW despite lower $[CO_{2(aq)}]$ (Figure 7). These trends contrast the global observed variability but are comparable to results from previous work in frontal regions (Bentaleb et al., 1998, Francois et al., 1993).

We predict the variability in $\varepsilon_p$ using temperature, $[CO_{2(aq)}]$, $\delta^{13}C_{CO2}$ and changes in community cell size across the region. If $[CO_{2(aq)}]$ was the controlling mechanism behind $\varepsilon_p$, increases in $[CO_{2(aq)}]$ would result in increased $\varepsilon_p$. There is no significant trend between modelled $\varepsilon_p$ (temperature, $[CO_{2(aq)}]$) and measured $\varepsilon_p$ (Figure 7a). In contrast, when cell size is included, there is a significant positive correlation (Figure 7c, r=0.72, p=0, df=18). These results indicate that in this region, there is an inverse trend between modelled $\varepsilon_p$ and $[CO_{2(aq)}]$: $\varepsilon_p$ increases with decreasing $[CO_{2(aq)}]$, which can be best attributed to the variability in the gross size structure of the phytoplankton assemblage across the SSTC (Figure 7).

If the flow of $[CO_{2(aq)}]$ into and out of a cell is determined by gas diffusion, then the flow is proportional to the cell surface area. A decrease in cell radius leads to an increase in cell surface area to volume ratio (SA:V), increasing the amount of $[CO_{2(aq)}]$ diffusing across the cell membrane relative to the total carbon within the cell, and allowing greater fractionation and higher $\varepsilon_p$. Thus $\varepsilon_p$ has been found to be negatively correlated to phytoplankton cell size, with larger cells such as diatoms showing less isotopic fractionation compared to smaller phytoplankton (Popp et al., 1998, Hansman and Sessions, 2016). These cell size trends are observed across our open ocean transect, with the largest cell sizes having lower $\varepsilon_p$ and higher $\delta^{13}C_{POC}$.

The influence of cell size on the expression of $\varepsilon_p$ is likely to have a greater effect with increasing growth rate (Rau et al., 1996, Popp et al., 1998). A higher growth rate increases the expression of a high $\varepsilon_p$ on smaller phytoplankton. The SSTC is a dynamic nutrient environment (Ito et al 2005), with the convergence of N limited subtropical waters (Eppley et al., 1979), with the iron limited ACC waters (Boyd et al., 2000, Browning et al., 2014). The convergence of contrasting regimes potentially increases nutrient availability to phytoplankton whilst also contributing to thermal stability of the upper water column (Longhurst, 1998), thus having the potential to elevate growth rates. Therefore the expression of cell size on $\varepsilon_p$ is intuitive in this environment. An increase in $\varepsilon_p$ with decreasing cell size has been noted in previous work (Goericke and Fry, 1994) and may be the primary driver for community $\varepsilon_p$ in the SSTC during spring and summer, when growth rates are high.

### 4.2 Regional and global factors influencing uptake fractionation

The changing $[CO_{2(aq)}]$ is the principal determinant of $\delta^{13}C_{POC}$ across the global ocean (Sackett et al., 1965; Rau et al., 1989; Goericke and Fry, 1994). A modelling study found that the inter-hemispheric differences in $\delta^{13}C_{POC}$ could be explained by the inter-hemispheric asymmetry in $[CO_{2(aq)}]$ (Hofmann et al., 2000). Poleward of ~50°S, $[CO_{2(aq)}]$ ranges between 15-25 µM,

$\delta^{13}C_{POC}$ between -30 and -24‰ and $\varepsilon_p$ is greatest of anywhere in the global ocean (Figure 8). The fractionation during carbon fixation (Rubisco) is highly expressed on $\varepsilon_p$ and other factors are less influential.

In the low latitude ocean, previous studies have shown that this trend becomes decoupled: $[CO_{2(aq)}]$ decreases, growth rates are more variable and community structure and seasonal dynamics decouple the observed correlation of fractionation with temperature (e.g. Francois et al., 1993, Bentaleb et al., 1998). Previous studies of $\varepsilon_p$ at the SSTC found decoupling between $\delta^{13}C_{POC}$ and $[CO_{2(aq)}]$, attributed to changing physical processes across the frontal region (Francois et al., 1993, Bentaleb et al., 1998). The variable water mass movements decouple trends and it has been suggested that $\delta^{13}C_{POC}$ variability in water masses can result from the local phytoplankton assemblage (Fontugne and Duplessy, 1978). Strong seasonal variations in $\delta^{13}C_{POC}$ can also result from changes in biological parameters such as cell radius, cell membrane permeability and growth rate (Francois et al., 1993; Goericke and Fry, 1994; Jasper et al., 1994; Laws et al., 1995; Popp et al., 1998). Phytoplankton assemblage-derived changes in $\varepsilon_p$ have been observed in other changing environments, such as the Seasonal Sea Ice Zone (Dehairs et al., 1997; Popp et al., 1999) and in major frontal regions (Dehairs et al., 1997; Popp et al., 1999; this study).

The results from our field study demonstrate that the phytoplankton assemblage has a key role in determining $\varepsilon_p$ and $\delta^{13}C_{POC}$ from their cell size and physiology, likely linked to the high growth rates in this frontal region. We test whether cell size variability presents a control over $\varepsilon_p$ across meridional transects (Figure 8). We find no relatable trend on a global scale and latitudinal trends demonstrate an increase in $\varepsilon_p$ with an increase in $[CO_{2(aq)}]$. However increased cell size reduces the expression of a high $\varepsilon_p$ (as shown with the higher $\delta^{13}C_{POC}$ and lower $\varepsilon_p$ in Figure 8c and d), which is particularly evident between 30-60°S. Thus, regions where frequent physical changes stimulate variable and diverse phytoplankton assemblages may be more likely to have a decoupled relationship between $\varepsilon_p$ and $[CO_{2(aq)}]$. We suggest that the high growth rates across this region play an important role in driving this change – we sampled across the SSTC in summer (high light levels), which may further promote the importance of cell size in determining $\delta^{13}C_{POC}$.

### 4.3 Changes to uptake fractionation and $\delta^{13}C_{POC}$ in response to climate change

Ambient $[CO_{2(aq)}]$ is increasing in the global ocean. A recent study found that $\varepsilon_p$ has increased significantly since the 1960s in the subtropical Atlantic, whereas no notable change has been detected in polar regions (Young et al., 2013). Our results suggest that a change in the community cell size would impact $\varepsilon_p$, with a decrease in cell radius leading to increased $\varepsilon_p$. Thus, a changing community structure with the onset of climate change may further impact the $\varepsilon_p$ and $\delta^{13}C_{POC}$, and have implications to our understanding of carbon isotope variability at the base of the food web.

Observational studies show rapid warming in the world's oceans in response to climate change, and that most of the ocean heat uptake is stored in the upper 75m (Cheng et al, 2019). Predicted ocean warming trends are variable in different regions, with a greater rate of increase predicted in the polar regions (IPCC, 2010). Warming at the ocean surface promotes thermal stratification, which, in the Southern Ocean may decrease light limitation, whereas in the subtropics is likely to promote further

nutrient limitation (Sarmiento et al., 2004). Thus, climate change will promote varying responses from phytoplankton communities and their physiology across the global ocean.

In the oligotrophic gyres, ocean-atmosphere GCMs project increased stratification and decreases in net primary productivity with the onset of climate change (Boyd and Doney, 2002; Capotondi et al., 2012; Le Quere et al., 2003). Nutrients are already the limiting factor for net primary productivity and small cells are readily adapted to these oligotrophic environments, where recycled nutrients such as ammonium are the main nutrients available to phytoplankton (Fawcett et al., 2011). Many studies observe a shift to phytoplankton communities dominated by picoplankton as the water column becomes stratified and increasingly nutrient depleted (Atkinson et al., 2003; Bouman et al., 2003; Latasa and Bidigare, 1998; Lindell and Post, 1995; Irwin and Oliver 2009). These observations suggest that the average community cell size may decrease further with ongoing climate change.

There may be large scale shifts in community structure, including the physiological status of phytoplankton and their ecological diversity (Bouman et al., 2005; Behrenfeld et al., 2005; Siegel et al., 2005). A decrease in cell size may lead to a faster pace of metabolism (Brown et al., 2004). However, recent work suggests that $CO_2$ fixation and respiration rates are unlikely to increase under nutrient limiting conditions (Maranon et al., 2018). Therefore subtropical regions may simultaneously experience warming, decreases in nutrient supply, increases in $CO_2$ availability, decreases in cell size and changes to community structure.

At higher latitudes, models predict increases in net primary production with improved light availability in the mixed layer and an extended growing season (Bopp et al., 2001, Sarmiento et al 2004). Warming and reduction in sea ice is likely to initiate an earlier onset of bloom with a predicted 5-10 days shift per decade (Henson et al., 2018). Therefore, in the subantarctic ocean we may expect decreased light limitation, higher growth rates and decreases in community cell size.

The results from this study demonstrate the different physiology of phytoplankton across the SSTC and the expression of carbon uptake on $\delta^{13}C$ fixed into the phytoplankton cell. We find that $\delta^{13}C_{POC}$ can be predicted using variability in cell size and $[CO_{2(aq)}]$. Further to this, the subtropical and subantarctic uptake fractionation may respond differently with changing ambient $[CO_{2(aq)}]$ and temperature with predicted future climate warming scenarios.

We use the results from this study to predict how the isotopic fractionation during carbon uptake may alter with increased $[CO_{2(aq)}]$ as a response to climate change. To do so, we alter the model inputs to increase atmospheric $CO_2$ from 400 ppm to 500 ppm, and thus increasing $[CO_{2(aq)}]$ in the surface ocean, which we calculate using the solubility coefficients of $CO_2$ in seawater (Figure 9, Weiss, 1974).

We test variability from an average cell radius of 5, 10 and 20 μm and investigate the changes over the temperature range of the ocean. Although a smaller cell size (radius 5 μm) fractionates $\delta^{13}C$ to a greater degree than a larger cell, there is only a 1‰ increase in $\varepsilon_p$ with a 100 ppm increase in atmospheric $CO_2$ concentrations (Figure 9a). Therefore changing $CO_2$ concentration alone may not have a large effect on $\delta^{13}C_{POC}$ in subtropical environments. Instead a trend to smaller average cell size would have a much greater impact on the $\delta^{13}C_{POC}$ which is observed and predicted in the oligotrophic gyres.

In the subantarctic waters, although predicted $\varepsilon_p$ is lower in the larger cell sizes expected south of the SSTC, an increase in ambient $CO_2$ concentrations would have a much larger effect on $\varepsilon_p$. There is an observed 3 ‰ increase in $\varepsilon_p$ in larger phytoplankton (radius 20 µm) with a 100 ppm increase in atmospheric $CO_2$ concentration. The changing conditions in the subantarctic ocean with the onset of climate change are also likely to promote the success of smaller sized phytoplankton, as the light conditions improve and the upper ocean becomes increasingly stratified in summer months (Bopp et al., 2003). A decrease in community cell size could also increase $\varepsilon_p$ and produce lower $\delta^{13}C_{POC}$. These results indicate that the subantarctic ocean, which has a relatively larger cell size in comparison to the subtropical ocean and is predicted to become increasingly stratified, may have a greater change in the $\delta^{13}C_{POC}$ produced during photosynthesis over the upcoming decades.

The sensitivity of $\delta^{13}C_{POC}$ to increases in anthropogenic carbon is determined by the change in $[CO_{2(aq)}]$ and also its isotopic signature. Enhanced diffusion of anthropogenic $CO_2$ between the atmosphere and the ocean's surface increases concentrations of $[CO_{2(aq)}]$ in the ocean (Friedli et al. 1986, Francey et al. 1999, Keeling et al. 2001). Anthropogenic $CO_2$ is enriched in the lighter $^{12}C$ isotope, so its invasion into the ocean decreases $\delta^{13}C_{DIC}$ in a phenomenon known as the Suess effect (Keeling, 1979), which has been observed across the ocean over the last decade (Quay et al. 2003). The uptake of anthropogenic $CO_2$ by the world's oceans has led to a decrease in $\delta^{13}C_{DIC}$ by 0.025% $yr^{-1}$ (Gruber et al., 1999). The increase in seawater $pCO_2$ from 400 to 500 µatm shown in Figure 9 corresponds to DIC increasing by from 30 to 50 µmol $kg^{-1}$ (with a greater DIC increase at higher temperatures). Assuming a ratio of anthropogenic $CO_2$ invasion to $\delta^{13}C_{DIC}$ change (i.e. ΔRC) of about −0.016 ‰ (µmol $kg^{-1})^{-1}$ in this region (Heimann and Maier-Reimer, 1996; McNeil et al., 2001), the associated Suess effect could decrease $\delta^{13}C_{DIC}$ – and therefore $\delta^{13}C_{POC}$ – by an extra 0.5 to 0.8 ‰, consistently across all cell sizes (Figure 9b). This decrease would be additional to, and independent from, any change due to fractionation, and consistent in magnitude for every cell size.

Seawater warming, which is expected to accompany future increases in $[CO_{2(aq)}]$, independently modulates the marine carbonate system (Humphreys, 2017) and the fractionation model of Rau et al. (1996). In this case, simultaneous warming would oppose the increase in $\varepsilon_p$ (and therefore decrease in $\delta^{13}C_{POC}$) driven by increasing $[CO_{2(aq)}]$, as shown by the negative line gradients in Figure 9a (and positive gradients in Figure 9b). However, this is expected to have a relatively small impact overall, as the following back-of-the-envelope calculation illustrates. Given an equilibrium climate sensitivity (i.e. the equilibrium warming of Earth's near-surface resulting from a doubling of atmospheric $pCO_2$) of 1.5 to 4.5 °C (Stocker et al., 2013), an increase in $pCO_2$ from 400 to 500 ppm would drive from 0.5 to 1.5 °C of global mean warming. For 10 µm cells, the $pCO_2$ change alone would increase $\varepsilon_p$ by ~1.8 ‰, while this warming alone would increase $\varepsilon_p$ by only 0.1 to 0.4 ‰, according to the model of Rau et al. (1996).

Stable isotope analysis of organic matter has emerged as the primary means for examining food web structure and variability. Carbon isotope signatures in particulate organic carbon vary substantially from the relative influence of terrestrial and marine carbon, carbon uptake pathways and the influence of carbon concentrating mechanisms (Jasper and Gagosian, 1990; Ganeshram et al., 1999). In contrast to nitrogen isotopes, there is negligible fractionation of carbon isotopes through trophic levels, which allows the accurate estimation of dietary sources of carbon. Therefore the factors which contribute to variability

at the base of the food web need to be understood well in order to accurately understand marine food web dynamics (Peterson and Fry, 1987).

This study highlights the importance of cell size as a primary determinant of the extent of isotopic fractionation in particulate organic carbon during uptake and the subsequent signature imparted at the base of the food web. Our findings support previous work predicting increases in $\varepsilon_p$ and decreases in $\delta^{13}C_{POC}$ in the future (Young et al., 2013). Yet we suggest that increasing $\varepsilon_p$ may predominantly result from shifts in community structure towards smaller sized phytoplankton. Our detailed study of the subtropical and subantarctic environments predict greater relative decreases in $\delta^{13}C_{POC}$ in polar regions than in the subtropics in response to changing $[CO_{2(aq)}]$. If increased stratification proceeds in the subantarctic, this may also lead to decreases in average cell size and thus even greater decreases in $\delta^{13}C_{POC}$.

## 5. Conclusions

$\delta^{13}C_{POC}$ measurements from the SSTC in the Atlantic Ocean are compared to model predictions to determine the factors which control $\delta^{13}C$ variability and carbon uptake fractionation ($\varepsilon_p$). Our results contrast global trends in marine waters, where $\delta^{13}C_{POC}$ is lower in high $CO_2$ environments as a result of increased carbon uptake fractionation. Instead we find the $\delta^{13}C_{POC}$ and $\varepsilon_p$ are largely determined by community cell size variability, which we estimate using phytoplankton pigment composition. We measured a greater $\varepsilon_p$ in the subtropical water masses where smaller sized phytoplankton are more dominant and can fractionate $\delta^{13}C$ to a greater degree by the increased $CO_2$ availability to the enzyme rubisco as a result of their enhanced increase surface area to volume ratio. Our results suggest a greater variability in $\delta^{13}C$ and $\varepsilon_p$ as a result of community cell size than previously predicted and highlight the need to understand the phytoplankton community structure.

We use our results from the field study to understand how increased $CO_2$ availability in the future will affect the carbon isotope fractionation in phytoplankton. Our findings suggest that larger celled phytoplankton in the subantarctic may respond more to changes in carbon concentration. However shifts in algal assemblages towards smaller phytoplankton will also have a large effect on the community $\varepsilon_p$ expressed. These results suggest that decreasing cell size and increased $CO_2$ availability to phytoplankton will increase $\varepsilon_p$ and decrease $\delta^{13}C_{POC}$. Coupled with further decreases in $\delta^{13}C_{POC}$ driven by the Suess effect, these factors will have implications for our use of $\delta^{13}C$ in food web studies in a changing marine environment.

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

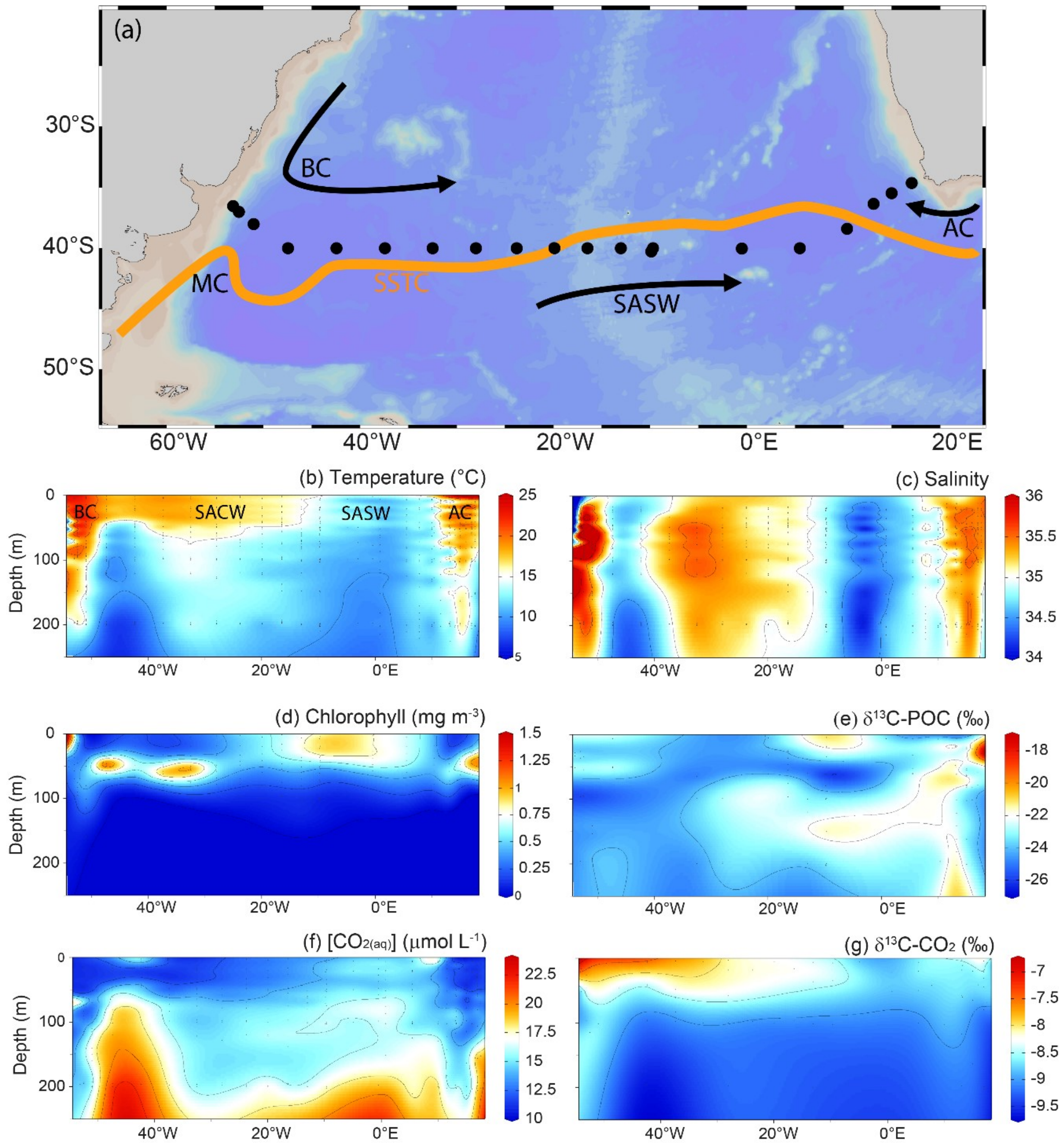

**Figure 1. Map and longitudinal transects across the south subtropical convergence.** (a) Map of study region, the orange line depicts the subtropical front (SST=16°C, from Browning et al., 2014). Longitudinal transects of (b) temperature, (c) Salinity, (d) Chlorophyll, (e) $\delta^{13}C_{POC}$, (f) $[CO_{2(aq)}]$, and (g) $\delta^{13}C_{CO2}$ in the upper 250 m. The water masses are identified in (b), BC=Brazil Current, SACW=South Atlantic Central Water, SASW=Subantarctic Surface Water, AC=Agulhas Current.

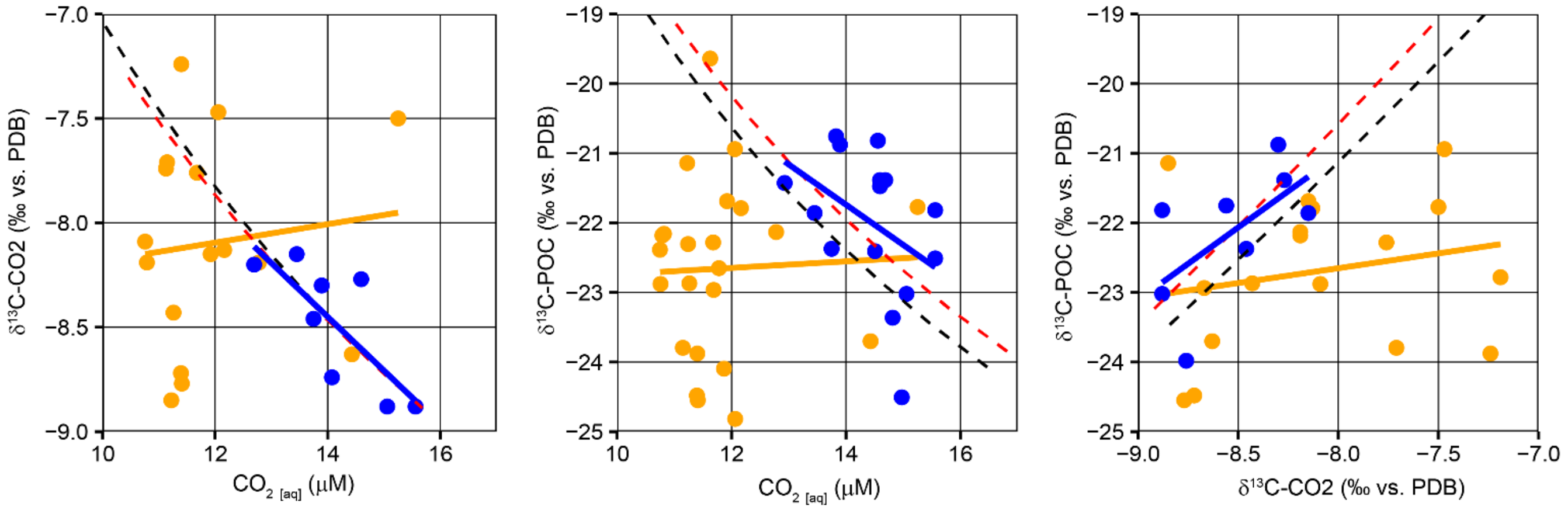

**Figure 2. Correlations between $[CO_{2(aq)}]$, $\delta^{13}C_{POC}$, and $\delta^{13}C_{CO2}$ in surface waters.** (a) $\delta^{13}C_{CO2}$ versus $[CO_{2(aq)}]$, (b) $\delta^{13}C_{POC}$ versus $[CO_{2(aq)}]$, (c) $\delta^{13}C_{POC}$ versus $\delta^{13}C_{CO2}$. Blue points represent SASW samples and orange points represent subtropical samples. A linear regression for each region is shown in the respective colour, all regressions were insignificant ($p>0.05$), apart from SASW samples in (a) where $r=-0.77$, $n=12$, $p=0.003$. The dashed line displays the expected trend due to diffusive uptake of carbon by phytoplankton using temperature and not cell size, red= atmospheric $CO_2$ 350ppm, 1.7 ‰, (Rau et al., 1996), black= 390ppm, 1.3 ‰ (representative of this study).

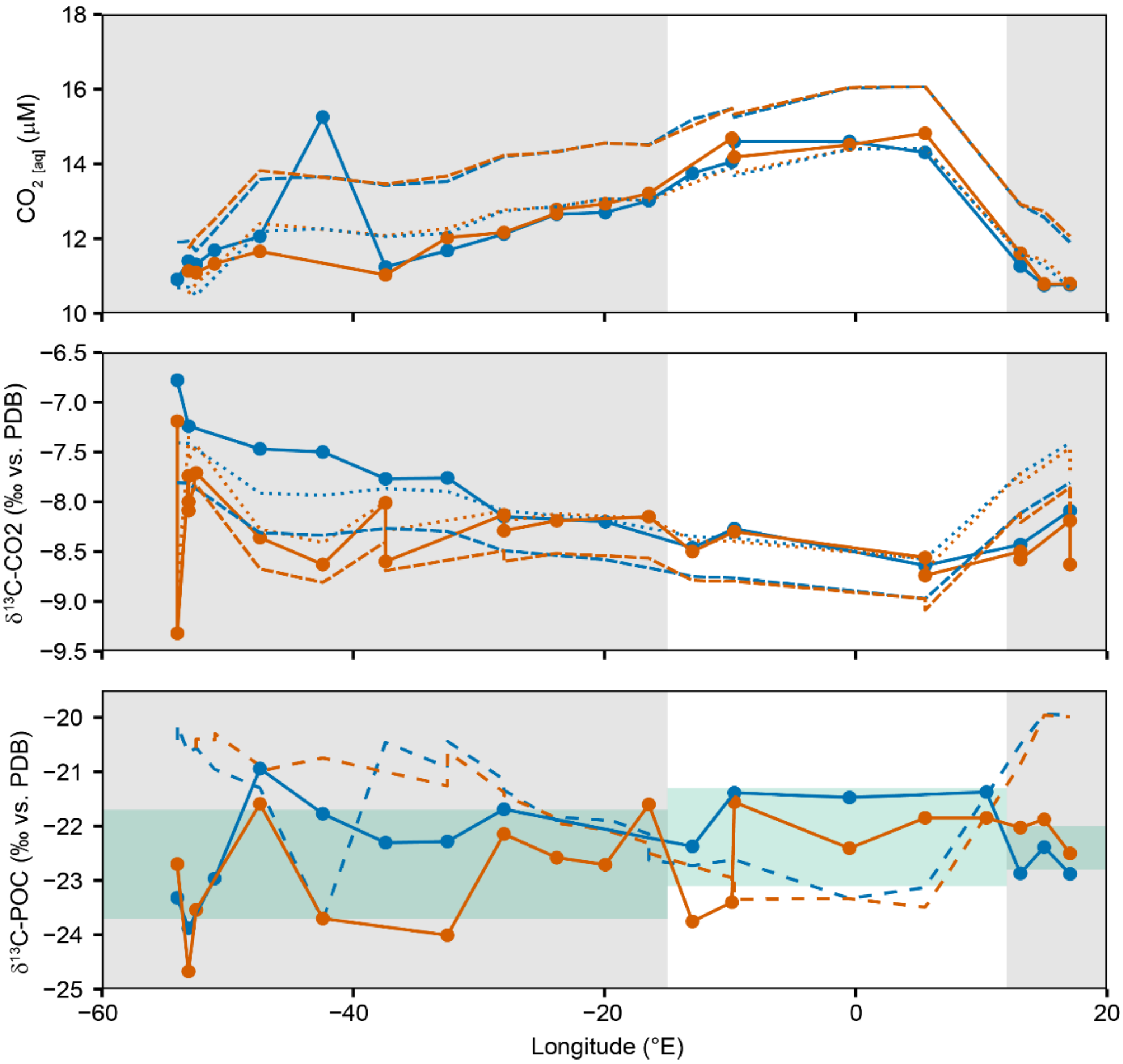

**Figure 3. Distribution of (a) [$CO_{2(aq)}$] (b) $\delta^{13}C_{CO2}$ and (c) $\delta^{13}C_{POC}$, versus longitude.** Closed circles and solid lines show measured values and trends. Orange lines and points = 5 m, Blue lines and points = 20 m. In (a) and (b), Dashed lines are modelled estimates using temperature only (Rau et al., 1996). Long dash: atmospheric $CO_2$ = 390ppm, dotted line atmospheric $CO_2$ =350ppm. In (c) Dashed lines are modelled estimates using temperature, [$CO_{2(aq)}$] and $\delta^{13}C_{CO2}$. Grey shaded areas highlight the stations sampled north of the SSTC. Green shaded bars represent 2σ for SACW, SASW and AC regions of the transect.

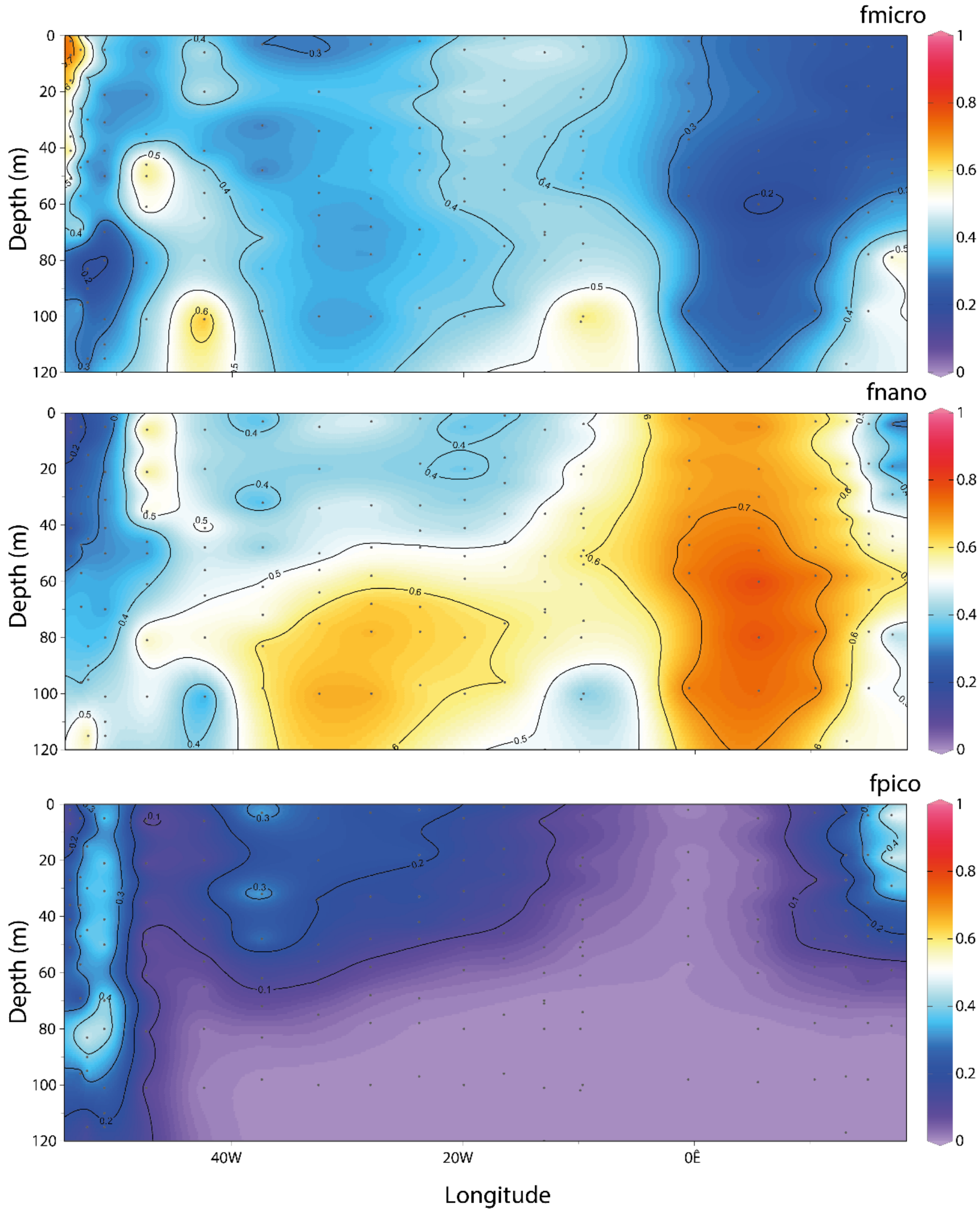

**Figure 4. The fractional contribution of phytoplankton size classes to total chlorophyll as estimated from phytoplankton pigments.** The size classes are defined as pico < 2 µm, nano = 2 – 20 µm and micro = 20 – 200 µm. 'f' signifies fractional contribution to chlorophyll-a concentration. Size class estimates were calculated following (Uitz et al., 2008).

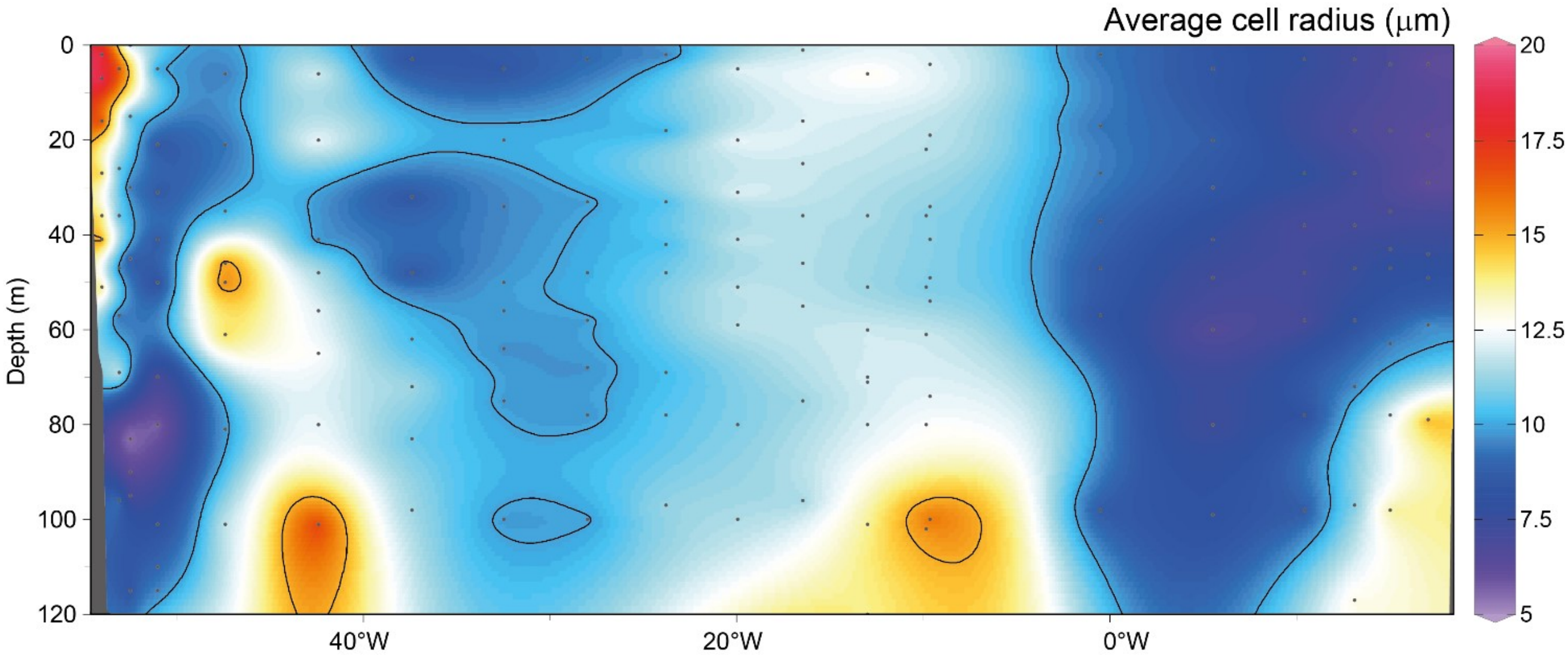

**Figure 5. The estimated average phytoplankton community cell radius.** The average radius was calculated using the proportions of pico-, nano- and microplankton in Figure 4. We estimate the average radius using assumed cellular radii of 0.5, 2.5 and 25 μm for pico-, nano- and microplankton, respectively.

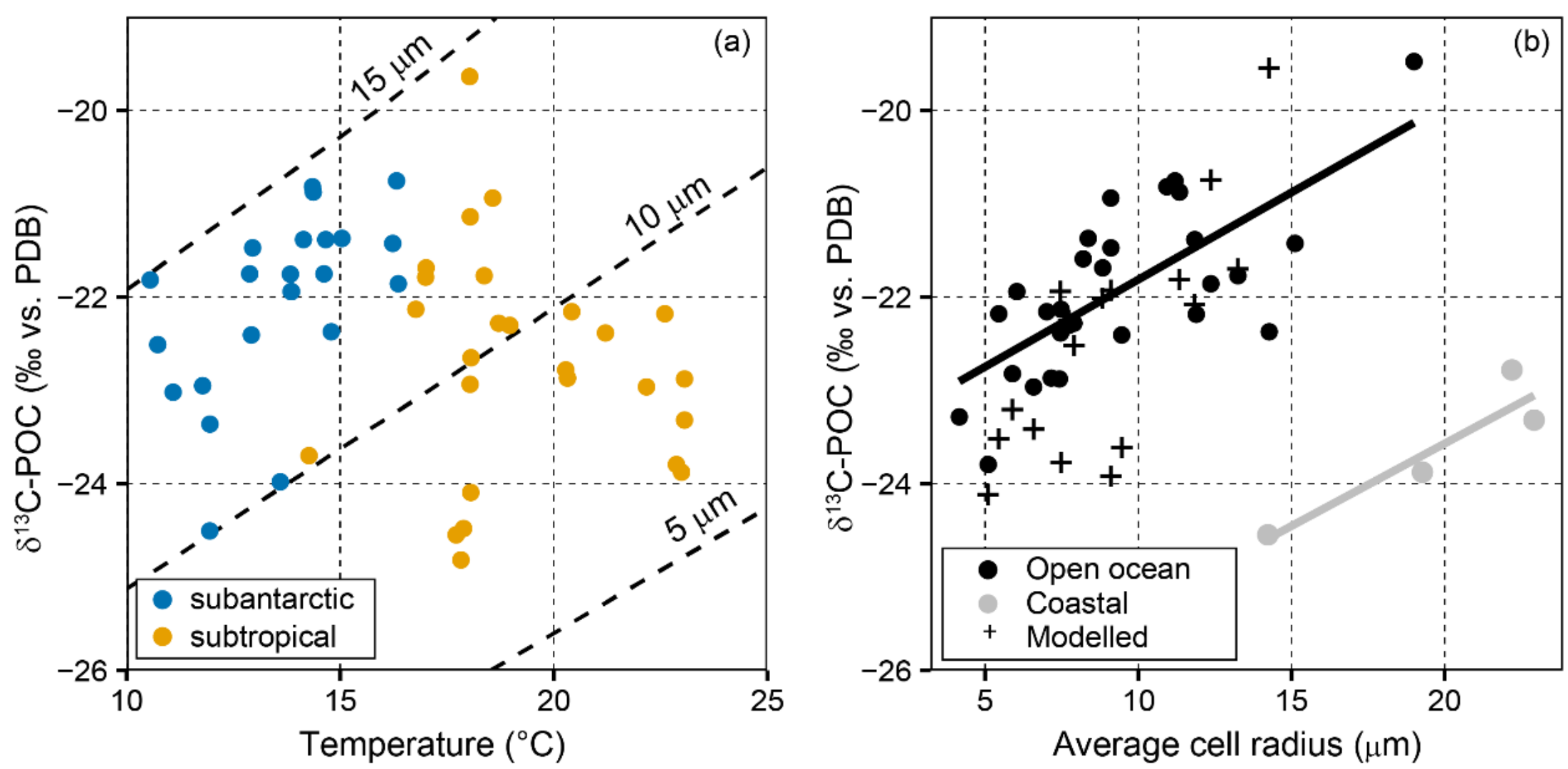

**Figure 6. $\delta^{13}C_{POC}$ variability and model predictions with temperature and cell size.** (a) $\delta^{13}C_{POC}$ versus temperature, with the modelled estimates for cell radii of 5, 10 and 15 μm. Blue circles = SASW, orange circles = subtropical waters. (b) $\delta^{13}C_{POC}$ vs cell radius as derived from pigment data. Samples from the eastern and western margin are excluded from correlation estimates. Rio Plata: r=0.92, df=2, p=0.075; Open ocean: r=0.72, df=30, p<0.001. Modelled $\delta^{13}C_{POC}$ is estimated using measured temperature and cell size and an assumed constant cell growth rate of 1.1 $d^{-1}$. Black crosses show modelled results. Average cell radius calculated from micro-plankton (25 μm), nano-plankton (2.5 μm) and pico-plankton (0.5 μm).

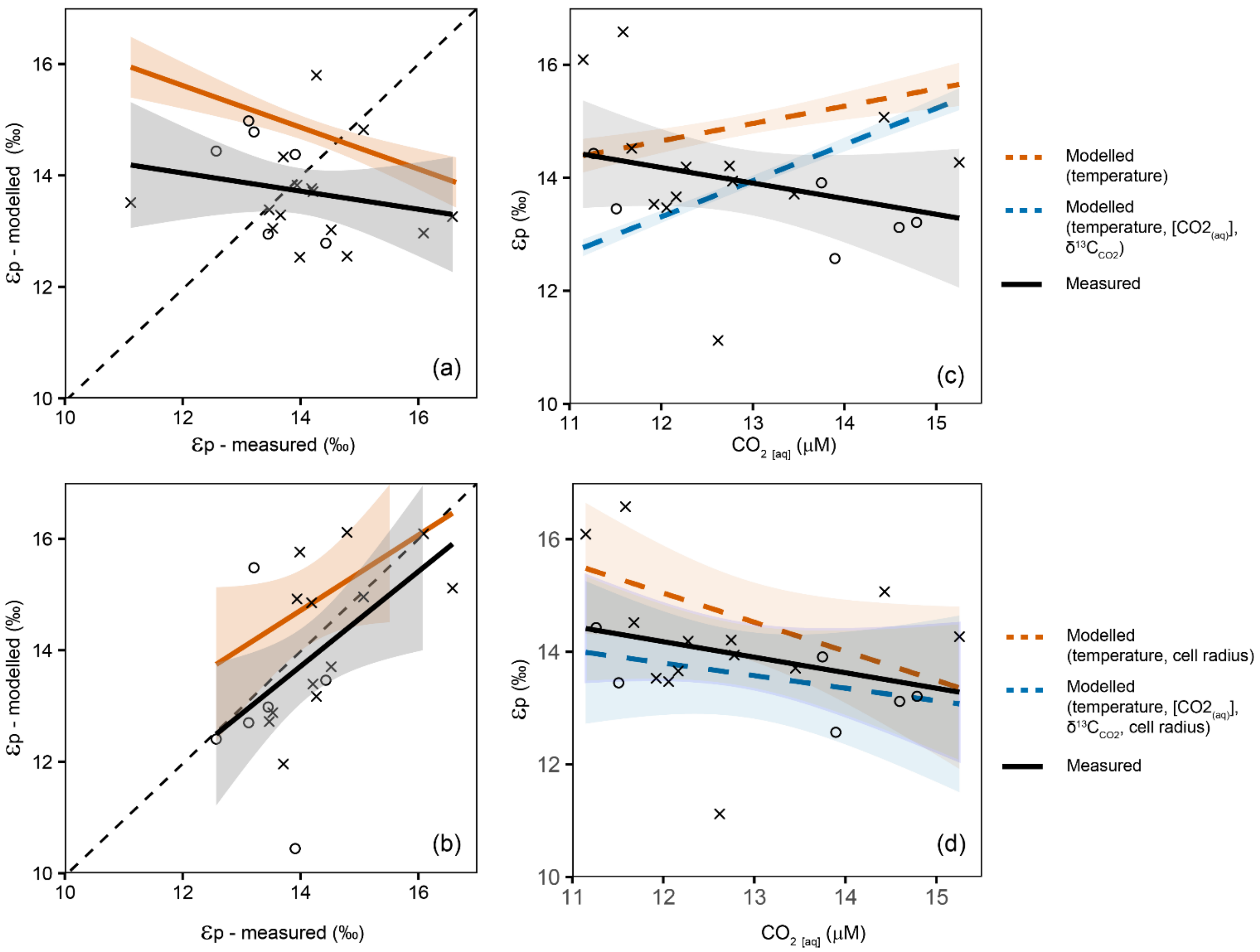

**Figure 7. Variation in modelled and measured $\varepsilon_p$ in the upper 60m with changing $[CO_{2(aq)}]$.** Black points and lines show measured $\varepsilon_p$, circles = SASW, crosses = subtropical water masses. (a) and (c) the regressions for modelled and measured $\varepsilon_p$, (b) and (d) modelled and measured $\varepsilon_p$ against $[CO_{2(aq)}]$. The top panel explores the predicted $\varepsilon_p$ using temperature, $[CO_{2(aq)}]$ and $\delta^{13}C_{CO2}$. The bottom panel explores the predicted $\varepsilon_p$ using temperature, $[CO_{2(aq)}]$, $\delta^{13}C_{CO2}$ and cell size.

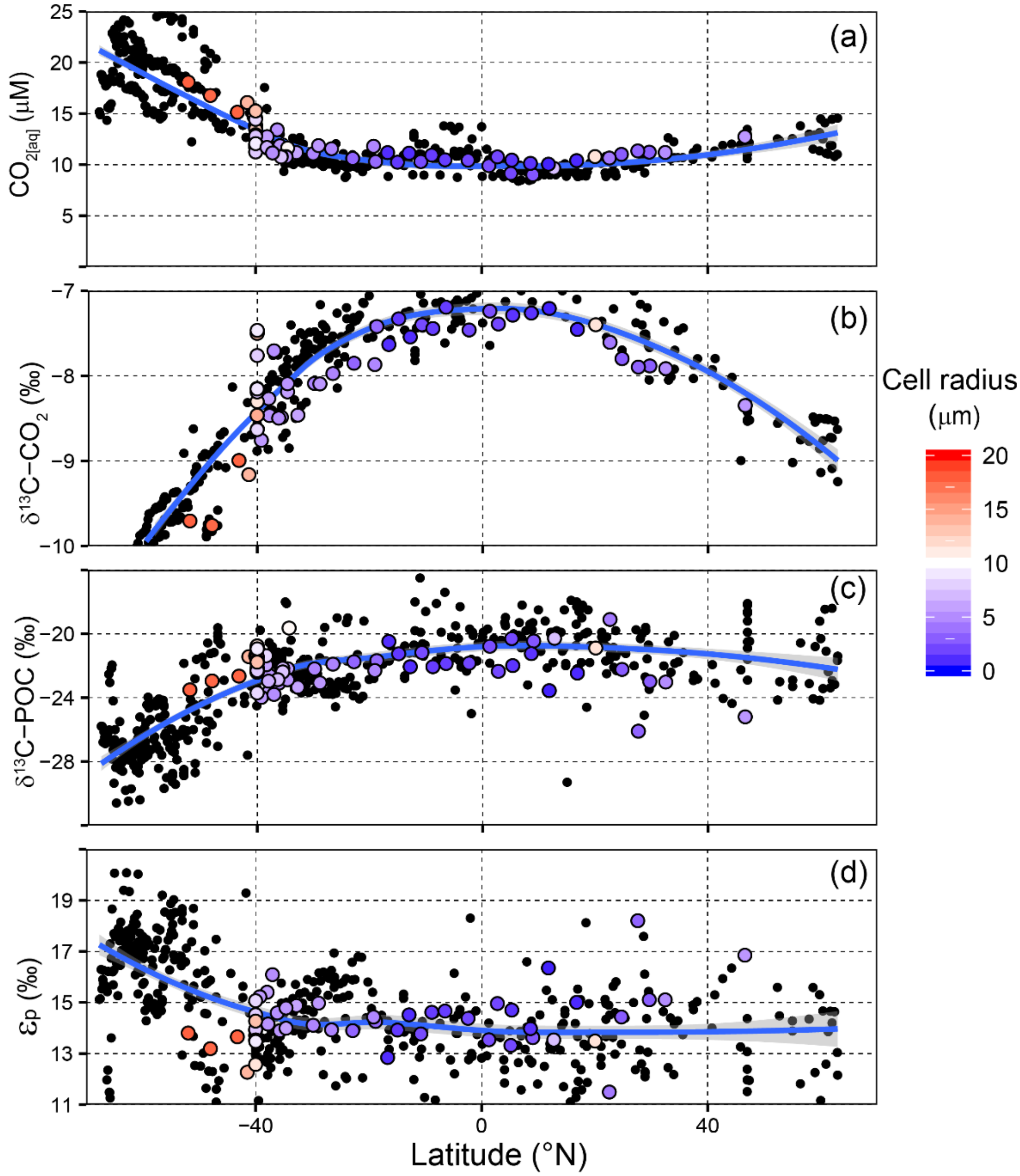

**Figure 8. Global distributions of (a) [$CO_{2(aq)}$], (b) $\delta^{13}C_{CO2}$, (c) $\delta^{13}C_{POC}$ and (d) $\varepsilon_p$ in surface waters, plotted against latitude.** Data include samples from $\delta^{13}C_{POC}$ compilation in Young et al., 2013 and data from this study. Coloured points show cell radius estimates (AMT3, AMT18 and this study). Blue lines in each show a loess fitted curve for the dataset.

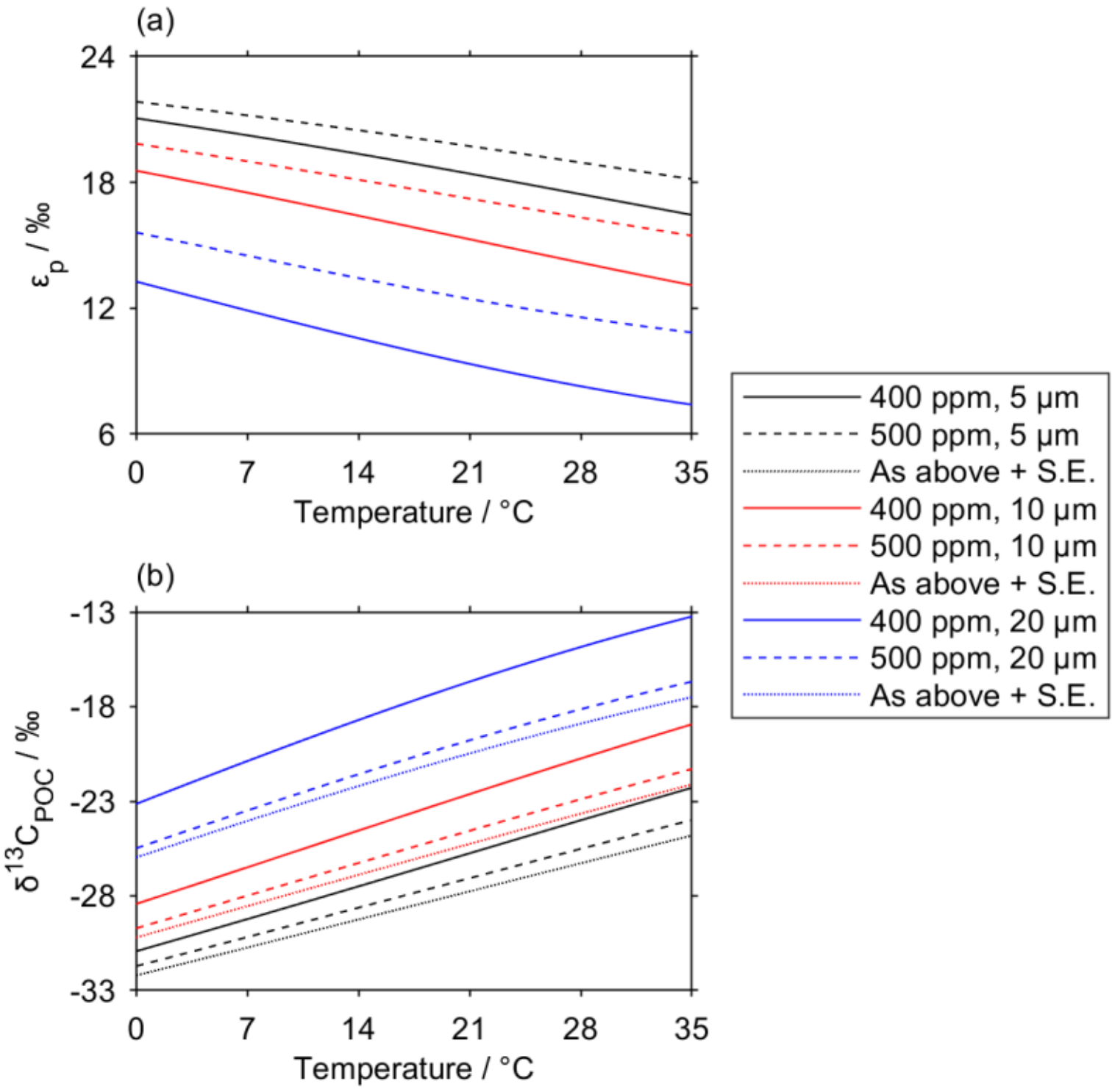

**Figure 9. Projected changes in $\varepsilon_p$ and $\delta^{13}C_{POC}$ due to ongoing anthropogenic emission and upper ocean uptake of $CO_2$ for different temperatures and cell sizes.** Variation in $\varepsilon_p$ with seawater temperature for $CO_2$ partial pressures of 400 ppm (solid line) and 500 ppm (dashed line) and average community cell radii of 5 µm (black), 10 µm (red) and 20 µm (blue). (b) Variation in $\delta^{13}C_{POC}$ under the same conditions as in (a), also showing the additional isotopic change driven by the Suess effect (labelled "S.E.", dotted lines). $CO_{2(aq)}$ is calculated using the atmospheric $CO_2$ concentration and the solubility of $CO_2$ in seawater (Weiss, 1974).