# Peer review of "Isotopic fractionation of carbon during uptake by phytoplankton across the South Atlantic subtropical convergence"

_Biogeosciences, 2019_

## Referee Comment (RC1) · Anonymous Referee #1 · 14 May 2019

This paper presents data about spatial variation the carbon isotopic composition of POC and DIC in the subtropical convergence zone of the south Atlantic and authors interpret their findings in terms of $CO_2$ solubility (temperature) and phytoplankton physiology (cell size; growth rate). Authors conclude to an important weight of cell size and growth rate in setting the measured isotopic composition of the phytoplankton. They also discuss the impact of the ongoing atmospheric $CO_2$ increase and resulting ocean warming on future isotopic fractionation and phytoplankton isotopic composition.

The authors are rather conservative in providing information about some methods used. In particular about the following: Phytoplankton size classes as deduced from

pigment assemblage. Just referring to Bricaud and Uitz seems hardly enough .. the few lines (27 to 33) at page 6 don't really enlighten this issue. The same holds for the use of Rau's diffusion model. There is no discussion whatsoever about 'why this model (which is quite complex) is selected, neither about the model parameters (including growth rate) which are mainly taken from the original Rau paper.

Page 6, Line 15 (and also page 7, line7): It is not clearly stated how model estimates based on 'temperature alone' are obtained, except for a reference to Rau et al. 1989. Is this the same as the original Farquhar model as described by François et al. ? If so, that model does not consider cell size .. but you mention a constant cell size of 10 $\mu$m was used. Please clarify.

Page 7, Line 1 and Figure 5: authors state that cell radii were smaller in the subtropical waters compared to the SASW. From Figure 5 this is hardly visible.

Page 8 Line 6: these trends 'contrast' the global observed variability. . . They contrast in what sense?

Page 8, line 21: the sentence 'A higher growth rate increases the expression of a high $\varepsilon$p on smaller phytoplankton' is unclear. Please reformulate.

Page 8 lines 20 to 27: the whole of the discussion here is highly hypothetical, and only yields a statement that waters north of the SSTC have 'the potential to elevate growth rates'. Later in the discussion it seems the 'potential for' has become a solid fact (e.g. line 14 and lines 19-20 at page 9).. Also it is likely that this frontal area is influenced by N-nutrient rich AAIW and SAMW waters. Have the authors considered this ?

Page 9, line20: Why would decreased light limitation lead to higher growth rates ? Higher biomass and higher primary production, yes, but why higher growth rates ?

Page 9, Lines 16-20: increase cell size reduces the expression of a high $\varepsilon$p as shown by the higher $\delta$13CPOC and lower $\varepsilon$p ..). It seems to me the data points rather fit the general trend of $\delta$13C and ep, and highlighted offset mentioned, appears weak.

Though this is not the subject of this paper, it is interesting to see this decrease of $\delta$13C DIC also in cold North Atlantic waters. What is the explanation for this phenomenon? Page 10, line 5: ".. predicting increases in $\varepsilon$p and decreases in $\delta$13CPOC " . Figure 9 rather shows increasing temperature would result in decreased $\varepsilon$p and increased $\delta$13CPOC ..

Page 12: the authors conclude to the significance of their findings for future studies of $\delta$13C in food web studies. They could add that this also extends to future studies about the fate of plankton organic matter in the deep ocean. In that aspect an useful paper that can be cited is the one by Cavagna et al., BG 10, 2013 "Water column distribution and carbon isotopic signal of cholesterol, brassicasterol and POC in the Atlantic sector of the S.O."

Minor things: Page 4 line 6: 50 ml of 100% HgCl2 were added; I guess you mean 50 $\mu$l ..? Page 10, line 11: the wording 'physiological status' is rather vague.. can you specify more ? Figures 4 and 5: mark the waters located north and south of the SSTC Figure 7: the full red line is not specified

---

## Referee Comment (RC2) · Anonymous Referee #2 · 28 May 2019

This is an interesting paper looking at the variability in carbon isotope (and fractionation) of particulate organic matter (with CO2aq) in relation to phytoplankton cell size. The authors sampled subantarctic and subtropical regimes with contrasting environments and community structures to investigate mechanisms for isotopic fractionation in d13CPOC resulting from carbon uptake and biological production in the upper ocean. The authors suggest that cell size is an important factor. Using estimates of cell size (via HPLC analyses) and calculated CO2aq, the authors suggest that smaller cells will respond less to increased CO2aq than the larger cells south of the SSTC and the wider Southern Ocean.

[Figure]

Query: when looking at investigating future epsilon-p did the authors consider the combined effect of increased CO2 and increased temperature in the two environments?

General point about Figures, it is very hard to deduce where measurements were taken in the profiles and also which interpolations were used to create the profiles.

Initial thoughts while starting to read the manuscripts were: 'but what about species composition'? This really only gets dealt with in the discussion. It would be good to see this upfront, including a small discussion about cell size on its own (so possibly discussing culture studies) actually supports what the authors conclude.

Introduction:

Second sentence: missing a bit; anthropogenic CO2 input to the atmosphere causes enhanced greenhouse gasses, which causes the oceans to warm up. It is not a direct effect.

Methods: A bit strange to see details of where the inorganic carbon isotopes where analysed, but none of the other analyses.

Results: 3.1 first para. In reference to Figure 1, what does MC stand for?

Figure 1 does not show a correlation between various variables, just cross sections.

3.2 Para 3 'There is no significant correlation between d13CPOC and CO2aq or d13CCO2 (Fig 2)' where? Subtropical samples?

Para 4 Statement: Picoplankton were dominant in the subtropical environments. NO. This figure suggests that fmicro and fnano are dominant in all environments.

The authors claim there is a significant positive correlation between average community cell radius and d13CPOC, with n=30. There are 47 data points in Figure 6a; in Figure 6b 4 are attributed to being coastal sites. What happened to the missing 13 data points?

Page 7: with to first sentence and reference to Figure 5: what is the average error and is the suggested difference supported by statistics?

Discussion: add some references when discussing the used of stable isotopes of organic matter as a primary means for examining food web structure and variability. Plus also to line 32-33 (nitrogen isotopes).

―――――――――――――――――――

---

## Author Comment (AC1) · 31 May 2019

We thank the reviewer for their time in completing this review, we believe that their input will help greatly improve the manuscript. Here we include responses to all of the comments: (1) Reviewer's comment (2) Author's comment (3) Suggested change to manuscript

(1) This paper presents data about spatial variation the carbon isotopic composition of POC and DIC in the subtropical convergence zone of the south Atlantic and authors interpret their findings in terms of CO2 solubility (temperature) and phytoplankton physiology (cell size; growth rate). Authors conclude to an important weight of cell size and
growth rate in setting the measured isotopic composition of the phytoplankton. They also discuss the impact of the ongoing atmospheric CO2 increase and resulting ocean warming on future isotopic fractionation and phytoplankton isotopic composition.

The authors are rather conservative in providing information about some methods used. In particular about the following: Phytoplankton size classes as deduced from pigment assemblage. Just referring to Bricaud and Uitz seems hardly enough .. the few lines (27 to 33) at page 6 don't really enlighten this issue. The same holds for the use of Rau's diffusion model. There is no discussion whatsoever about 'why this model (which is quite complex) is selected, neither about the model parameters (including growth rate) which are mainly taken from the original Rau paper.

(2) We thank the reviewer for raising the concern that there is not enough information for the reader regarding the use of size classes and also the use of the Rau, 1996 model. We now include further information below and we suggest that this information could be included in the supplementary information for the manuscript:

Size class calculations

The size classes of phytoplankton were calculated using seven diagnostic pigments which are used as biomarkers of specific taxa as calculated from the HPLC data (see methods). The taxa can be used to estimate the proportion of micro, nano and pico-phytoplankton. This is calculated using the following formulae:

wDP=1.4(fucoxanthin) +1.41(peridinin) + 0.60(alloxanthin) + (0.35('19 –BF) + 1.27(19'-HF) + 0.86(zeaxanthin) +1.01(Chl b + divinyl - Chl b)

fmicro = (1.41(fucoxanthin) 1.41(peridinin)/wDP

fnano = (0.60(alloxanthin) 0.35(19' - BF) 1.27(19' - HF)/wDP

fpico = (0.86(zeaxanthin) 1.01(Chl b) divinyl - Chl b)/wDP

The coefficients represent the average ratio between chla and the concentration of

each diagnostic pigment, which are broadly related to taxa. This method contains caveats, which include: -pigments are shared across taxa

-cells adjust their pigments ratios in response to light/nutrient stress

-this proxy was derived for a global study to estimate phytoplankton groups from satellites, therefore, the shifts in size structure as you go from the gyres (Prochlorococcus dominated) to an upwelling system (diatom dominated) are nicely captured but the high latitudes are misrepresented.

In this dataset we transition from gyre-like to mesotrophic conditions, which we believe should be accounted for relatively accurately with this method. Bricaud et al., 2004 also found a good correspondence to the optical properties of phytoplankton, which can be viewed as an independent proxy of cell size.

Rau et al., 1996 model

On initial experiments for this work, it was found that [CO2(aq)] alone was not a suitable determinant of the d13C of POC in surface waters across the SSTC, therefore the importance of other factors needed to be examined. The Rau model is used as the intracellular carbon concentration is dependent on [CO2(aq)], cell radius, cell growth rate, cell membrane permeability to [CO2(aq)] and temperature. This therefore allows the importance of these variables to be tested.

We include a link to the MATLAB code for the model: https://github.com/mvdh7/miscellanea/blob/master/g40s_isotopes/rau1996.m

(1) Page 6, Line 15 (and also page 7, line7): It is not clearly stated how model estimates based on 'temperature alone' are obtained, except for a reference to Rau et al. 1989. Is this the same as the original Farquhar model as described by Francois et al. ? If so, that model does not consider cell size .. but you mention a constant cell size of 10 $\mu$m was used. Please clarify.

(2) Using temperature alone signifies that all other baseline numbers within the model

construct from Rau et al., 1996 have been used apart from the variability in temperature across the SSTC (see above table). The temperature is used to reconstruct CO2, which is used to predict variability in d13CPOC. All other variables within the model construct (see previous comment), are used as constants from the Rau model, therefore a constant cell size and growth rate.

(3) To investigate the spatial variability across the SSTC, [CO2(aq)], and $\delta$13CCO2 were plotted against longitude and compared to model estimates (Rau et al., 1996, supplementary information), where we used the model constants for cell size (10um) and reconstructing [CO2(aq)] from temperature variability across the transect (Figure 3a and b).

(1) Page 7, Line 1 and Figure 5: authors state that cell radii were smaller in the sub-tropical waters compared to the SASW. From Figure 5 this is hardly visible.

(2) We thank the reviewer for highlighting this point, we agree that the colour plot makes it challenging to identify where the change in cell radii matches with the change in water mass. We attach a suggested amended figure with isotherms overlain for temperatures of 14 and 18 °C (Figure AC1). The smallest cell radii are within the core of the Agulhas and Brazil currents (where cell radii <8um and temperature >18°C). The largest cell radii at the surface (excluding the Rio Plata) are within the SASW. Following on from a further comment below, we have also highlighted where stations were relative to the SSTC (north or south).

There is a significant moderate negative correlation between average cell radius and salinity (Figure AC2, Pearson's product moment correlation: r=-0.56, t = -8.69, df = 165, p-value = 3.36e-15). The lowest salinities from the Rio Plata have been removed from this (<33).

(1) Page 8 Line 6: these trends 'contrast' the global observed variability. . . They contrast in what sense?

(2) This is an important point and we recognise that our wording needs to be clearer within the revised manuscript. This comment within the manuscript is describing the global trends: when CO2aq is high, $\varepsilon$p is high (i.e. in the Southern Ocean) and in areas where [CO2(aq)] is low such as the subtropical gyres, $\varepsilon$p is lower (see Figure 8). In our dataset the lowest ep is where the [CO2(aq)] is the highest (and cell size is larger), see red/pink points in Figure 8a and d.

Suggested new wording: (3) Our data contrast the global observed variability (of high $\varepsilon$p in high [CO2(aq)] regions such as the Southern Ocean) but are comparable to results from previous work in frontal regions where higher $\varepsilon$p has been observed in lower [CO2(aq)] subtropical water masses (Bentaleb et al., 1998, Francois et al., 1993).

(1) Page 8, line 21: the sentence 'A higher growth rate increases the expression of a high ep on smaller phytoplankton' is unclear. Please reformulate.

(2) In Rau's 1996 model – when growth rate is higher, the effect of a variable surface area to volume ratio (or cell size) is expressed more on $\delta$13CPOC. A higher growth rate increases the expression of a low $\varepsilon$p on larger phytoplankton compared to lower growth rates (Fry and Wainwright, 1991). Wording changed to:

(3) A higher growth rate, such as in spring/summer blooms increases the range in $\varepsilon$p expressed across cell sizes. For instance in fast growing blooms, larger cell sizes may have higher relative d13CPOC and lower $\varepsilon$p than smaller cell sizes, compared to in low growth periods (e.g. Fry and Wainwright, 1991).

(1) Page 8 lines 20 to 27: the whole of the discussion here is highly hypothetical, and only yields a statement that waters north of the SSTC have 'the potential to elevate growth rates'. Later in the discussion it seems the 'potential for' has become a solid fact (e.g. line 14 and lines 19-20 at page 9).. Also it is likely that this frontal area is influenced by N-nutrient rich AAIW and SAMW waters. Have the authors considered this ? Page 9, line20: Why would decreased light limitation lead to higher growth rates? Higher biomass and higher primary production, yes, but why higher growth rates?

[Figure]

(2) The nutrient rich SAMW (500m) and AAIW (750m) waters are deeper in the water column here, but the SASW originates from the surface waters of the polar frontal zone (ultimately sourced from the UCDW) and the northwards flowing waters which have high N in comparison to the subtropical waters (Tuerena et al., 2015). The SSTC creates an environment where there is the convergence of N-limited subtropical waters and Fe-limited subantarctic waters (Browning et al., 2014). This region therefore has the potential to alleviate nutrient stress. The convergence of water masses at the SSTC can also lead to strong and swift stratification and alleviation of light limitation, which would lead to higher growth rates (Llido et al., 2005). A study of the SSTC south of New Zealand found growth rates more than double the rates within the sub Antarctic and subtropical water masses (Delizo et al., 2007). We suggest that over the broader region, growth rates here will be higher across the SSTC than in the South Atlantic gyre or in the Southern Ocean.

(1) Page 9, Lines 16-20: increase cell size reduces the expression of a high ep as shown by the higher $\delta13CPOC$ and lower $\varepsilon p$ ..). It seems to me the data points rather fit the general trend of $\delta13C$ and ep, and highlighted offset mentioned, appears weak.

(2) The data points fit the trend to a lesser degree than expected for [CO2(aq)]. Note the red-pink points in 8a have a higher than predicted [CO2(aq)] at the given latitude (40-50°S) and a lower than predicted $\delta13CCO2$. It would be intuitive therefore to predict that the $\delta13CPOC$ produced would be lower than the average trend, but is in fact higher. $\varepsilon p$ is also lower than predicted with all of the cell radii >10um.

(1) Though this is not the subject of this paper, it is interesting to see this decrease of $\delta13C$ DIC also in cold North Atlantic waters. What is the explanation for this phenomenon? The lower $\delta13C$ in the North Atlantic and Southern Ocean is related to circulation and the relative extent of photosynthesis and respiration of nutrients and carbon within surface waters. In the low latitude ocean, nutrients and DIC are much lower in the surface ocean from downwelling and the uptake of nutrients and regeneration at depth, therefore $\delta13CDIC$ is higher in the lower latitudes compared to the higher

latitudes. The concentrations of DIC and nutrients are higher in the Southern Ocean compared the North Atlantic (more upwelling), therefore the $\delta13CPOC$ is lower relative to the North Atlantic.

(1) Page 10, line 5: ".. predicting increases in $\varepsilon p$ and decreases in $\delta13CPOC$ " . Figure 9 rather shows increasing temperature would result in decreased $\varepsilon p$ and increased $\delta13CPOC$ .. On page 12, line 5.

Although an increase in temperature in the figure shows an increase in $\delta13CPOC$ and a decrease in ïĄěp, this will have very little effect compared to the predicted changes in carbon availability and cell size. To give an example:

A 2°C change in SST from 14 to 16°C would increase $\delta13CPOC$ from -23.9‰ to -23.3‰That is the predicted change over ∼200yrs (IPCC). Over this time period atmospheric CO2 would increase from pre-industrial to 500ppm which would decrease $\delta13CPOC$ to -26‰ (at 14°C) and -25.5‰ (at 16°C). Decreasing cell radius from 10um to 8um would decrease $\delta13CPOC$ further to -27‰ (14°C) and -26.5‰ (16°C).

Therefore a 2°C increase in SST with the expected rise in atmospheric CO2 would decrease $\delta13CPOC$ from -23.9‰ to -25.5‰ and would decrease further if the average cell size decreased.

(1) Page 12: the authors conclude to the significance of their findings for future studies of $\delta13C$ in food web studies. They could add that this also extends to future studies about the fate of plankton organic matter in the deep ocean. In that aspect an useful paper that can be cited is the one by Cavagna et al., BG 10, 2013 "Water column distribution and carbon isotopic signal of cholesterol, brassicasterol and POC in the Atlantic sector of the S.O."

(2)We thank the reviewer for this useful addition, and we will include this line of thought in the amended manuscript.

Minor things: (1) Page 4 line 6: 50 ml of 100% HgCl2 were added; I guess you mean

50 $\mu$l ..? (2) Correct, this has been amended (3) and 50 $\mu$L of 100% HgCl2 added

(1) Page 10, line 11: the wording 'physiological status' is rather vague.. can you specify more ?

Changed to: (3) including the physiological dependencies of phytoplankton on light and nutrients and their ecological diversity

(1) Figures 4 and 5: mark the waters located north and south of the SSTC (2) Now amended: green triangles have been added to the figures to mark the SSTC (see Figure AC1 and AC3).

(1) Figure 7: the full red line is not specified (2) See edited figure (Figure AC4)
* * *
[Figure]

**Fig. 1.** Figure AC1. Average cell radius across the SSTC (white contours show cell radius of 8 and 10μm). Black contours show temperatures of 14 and 18 °C. Green triangles mark the subtropical front.

**Fig. 2.** Figure AC2. Correlation between average cell radius and salinity, with temperature as colour.

[Figure]

**Fig. 3.** Figure AC3.

**Fig. 4.** Figure AC4.

[Figure]

---

## Referee Comment (RC3) · Anonymous Referee #1 · 11 Jun 2019

1/ A further comment conecerning the following reply of the authors (page C7):

"Although an increase in temperature in the figure shows an increase in $\delta$13CPOC and a decrease in ep, this will have very little effect compared to the predicted changes in carbon availability and cell size. "

I suggest authors make this future change (decrease) in d13CPOC more visible to the reader by marking it in Figure 9b. For example they could mark the jump from the 400 ppm to the 500ppm level with inceasing temperature by an arrow.

2/ In their reply on the question about the latitudinal distribution of d13C-DIC, the authors don't really clarify the issue, I believe. Of course Southern Ocean d13C-DIC is

very low because of upwelling of deep ocean waters depleted in 13C-DIC there, a phenomenon not present in the North Atlantic. So I feel the question about which process really imposes lower d13C-DIC in the North Atlantic is not satisfactorily resolved by their reply. Admitedly this is not the subject of their paper.

―――――――――――――――――

---

## Author Comment (AC2) · 3 Jul 2019

Response to reviewer 2. We thank the reviewer for their time in completing this review, we believe that their input will help greatly improve the manuscript. Here we include responses to all of the comments: (1) Reviewer's comment (2) Author's comment (3) Suggested change to manuscript

(1) This is an interesting paper looking at the variability in carbon isotope (and fractionation) of particulate organic matter (with CO2aq) in relation to phytoplankton cell size. The authors sampled subantarctic and subtropical regimes with contrasting environments and community structures to investigate mechanisms for isotopic fractionation in

d13CPOC resulting from carbon uptake and biological production in the upper ocean. The authors suggest that cell size is an important factor. Using estimates of cell size (via HPLC analyses) and calculated CO2aq, the authors suggest that smaller cells will respond less to increased CO2aq than the larger cells south of the SSTC and the wider Southern Ocean.

Query: when looking at investigating future epsilon-p did the authors consider the combined effect of increased CO2 and increased temperature in the two environments?

(2) We refer to our response to reviewer 1, which describes the expected changes to temperature as well as CO2 increases (the temperature increases would have a much lesser effect than CO2 increases and a decrease in cell size).

'Although an increase in temperature in the Figure 9 shows an increase in $\delta$13CPOC and a decrease in ep, this will have very little effect compared to the predicted changes in carbon availability and cell size. To give an example:

A 2degC change in SST from 14 to 16degC would increase $\delta$13CPOC from -23.9‰ to - 23.3‰ which is the predicted change over 200yrs (IPCC). Over this time period atmospheric CO2 would increase from pre-industrial to 500ppm which would decrease $\delta$13CPOC to -26‰ (at 14degC) and -25.5‰ (at 16degC). Decreasing cell radius from 10um to 8um would decrease $\delta$13CPOC further to -27‰ (14degC) and -26.5‰ (16degC).

Therefore a 2degC increase in SST with the expected rise in atmospheric CO2 would decrease $\delta$13CPOC from -23.9‰ to -25.5‰ and would decrease further if the average cell size decreased.'

In the revised version, we will include a statement to address the effects of both CO2 and temperature.

(1) General point about Figures, it is very hard to deduce where measurements were taken in the profiles and also which interpolations were used to create the profiles.
(2) We have edited Figures 1 and 4 to have more visible points in the profiles (larger point size). The interpolation for these figures has been made using ODV and the weighted average gridding (x, y spacing determined by profile spacing). Information about this has now been included into the captions.

(1) Initial thoughts while starting to read the manuscripts were: 'but what about species composition'? This really only gets dealt with in the discussion. It would be good to see this upfront, including a small discussion about cell size on its own (so possibly discussing culture studies) actually supports what the authors conclude.

(2) We will provide a paragraph in the manuscript introduction regarding species composition and we will refer to Figures 4 and 5 in Browning et al., 2014, which include the contribution of major accessory pigments to total accessory pigments.

(1) Introduction: Second sentence: missing a bit; anthropogenic CO2 input to the atmosphere causes enhanced greenhouse gasses, which causes the oceans to warm up. It is not a direct effect.

(2) Sentence changed to:

(3) Anthropogenic carbon inputs and the increase of greenhouse gases in the atmosphere are causing ocean warming (Cheng et al., 2019), changes to upper ocean stratification (Bopp et al., 2001; Capotondi et al., 2012) and altered distributions of nutrients and carbon (Khatiwala et al., 2013; Quay et al., 2003; Gruber et al., 2019).

(1) Methods: A bit strange to see details of where the inorganic carbon isotopes where analysed, but none of the other analyses.

(2) We agree with the reviewer and have removed 'University of Cambridge' from the manuscript. Sentence now reads: (3) Samples were measured using a Thermo MAT253 stable isotope mass spectrometer.

(1) Results: 3.1 first para. In reference to Figure 1, what does MC stand for?

(2) Sentence changed to: (3) The three subtropical water masses (Agulhas Current (AC), South Atlantic Central Water (SACW) and Brazil Current (BC)) can be readily identified with warmer temperatures and higher salinities, the influence of the Malvinas Current (MC) separates the core of the SACW and BC (Figure 1).

(1) Figure 1 does not show a correlation between various variables, just cross sections.

(2) Sentence has been edited to read:

(3) Across the zonal transect, higher $\delta$13CCO2 is associated with lower [CO2(aq)] and warmer temperatures of the subtropical water masses (Figure 1).

(1) 3.2 Para 3 'There is no significant correlation between d13CPOC and CO2aq or d13CCO2 (Fig 2)' where? Subtropical samples?

(2) Sentence has been edited to read:

(3) There are no significant correlations between $\delta$13CPOC and [CO2(aq)] or $\delta$13CCO2 in the subtropical or subantarctic water masses (Figure 2, p>0.05).

(1) Para 4 Statement: Picoplankton were dominant in the subtropical environments. NO. This figure suggests that fmicro and fnano are dominant in all environments.

(2) We thank the reviewer for highlighting this error in our wording and have changed the sentence accordingly:

(3) Picophytoplankton were more abundant in the subtropical environments in comparison to the SASW, contributing between 30-40% of the pigment biomass at the core of these water masses (Figure 4).

(1) The authors claim there is a significant positive correlation between average community cell radius and d13CPOC, with n=30. There are 47 data points in Figure 6a; in Figure 6b 4 are attributed to being coastal sites. What happened to the missing 13 data points?

(2) There is less data in Figure 6b as we did not have corresponding cell size data for all of the d13C-POC data points, to inform the reader, this information has been added to the figure caption.

(1) Page 7: with to first sentence and reference to Figure 5: what is the average error and is the suggested difference supported by statistics?

(2) In general there is no significant difference between the two water masses when you take the definitions of >14 and <14C for subtropical and subantarctic (south and north of the SSTC), as there is the convergence and mixing of water masses in this region. The large errors associated with the average cell radii can arise from the variation at the DCM of the subtropical water masses (larger size cells) and the variability from the mixing of water masses and thus different nutrient requirements. If we use only the cores of each of the surface water masses and discount the variability at the DCM, then there is a significant difference (Subtropical >20C 6.5 ±0.8, n17, Subantarctic<18C 10.4 ±2.3 n31 ). Because of this ambiguity, we change the wording accordingly:

(3) Estimated average cell radii were generally smaller at the core of the subtropical water masses compared to the SASW (Figure 5) (depth range <40m, subtropical >20C 6.5um ±0.8, n17, subantarctic<18C 10.4um ±2.3 n31).

(1) Discussion: add some references when discussing the used of stable isotopes of organic matter as a primary means for examining food web structure and variability. Plus also to line 32-33 (nitrogen isotopes).

(2) This is a valuable comment and extra references will be added to the revised version of the manuscript.

---

## Author Comment (AC3) · 3 Jul 2019

Here we include responses to all of the comments: (1) Reviewer's comment (2) Author's comment

(1) 1/ A further comment concerning the following reply of the authors (page C7): "Although an increase in temperature in the ïn ËŻAgure shows an increase in ËĞ $\delta$13CPOC and a decrease in ep, this will have very little effect compared to the predicted changes in carbon availability and cell size. "

I suggest authors make this future change (decrease) in d13CPOC more visible to the

reader by marking it in Figure 9b. For example they could mark the jump from the 400 ppm to the 500ppm level with inceasing temperature by an arrow.

(2) Agreed, we will amend Figure 9b accordingly.

(1) 2/ In their reply on the question about the latitudinal distribution of d13C-DIC, the authors don't really clarify the issue, I believe. Of course Southern Ocean d13C-DIC is very low because of upwelling of deep ocean waters depleted in 13C-DIC there, a phenomenon not present in the North Atlantic. So I feel the question about which process really imposes lower d13C-DIC in the North Atlantic is not satisfactorily resolved by their reply. Admitedly this is not the subject of their paper.

(2) We include a Figure to show the relationship between d13C-CO2 and CO2aq globally, using the data from Figure 8. The d13C falls in line with expected values for the given CO2aq of the North Atlantic (-9‰ red points in Figure1c). The Southern Ocean values are lower than the North Atlantic due to upwelling (-10-11‰, we will expand the axes in Figure 8b to make it more apparent (North Atlantic not as low as Southern Ocean).

Figure 1, relationship between d13C-CO2 and CO2aq. (a) map of data points, (b) d13C-CO2 and CO2aq with longitude as a z variable, (c) d13C-CO2 and CO2aq with latitude as a z variable.

Fig. 1.

---

## Author Response (AR1)

We thank the reviewer for their time in completing this review, we believe that their input will help greatly improve the manuscript. Here we include responses to all of the comments:
**(1) Reviewer's comment**
(2) Author's comment
*(3) Change to manuscript*

Response to reviewer 1.

**(1)       This paper presents data about spatial variation the carbon isotopic composition of POC and DIC in the subtropical convergence zone of the south Atlantic and authors interpret their findings in terms of CO2 solubility (temperature) and phytoplankton physiology (cell size; growth rate). Authors conclude to an important weight of cell size and growth rate in setting the measured isotopic composition of the phytoplankton. They also discuss the impact of the ongoing atmospheric CO2 increase and resulting ocean warming on future isotopic fractionation and phytoplankton isotopic composition.**
**The authors are rather conservative in providing information about some methods used. In particular about the following: Phytoplankton size classes as deduced from pigment assemblage. Just referring to Bricaud and Uitz seems hardly enough .. the few lines (27 to 33) at page 6 don't really enlighten this issue. The same holds for the use of Rau's diffusion model. There is no discussion whatsoever about 'why this model (which is quite complex) is selected, neither about the model parameters (including growth rate) which are mainly taken from the original Rau paper.**
(2)       We thank the reviewer for raising the concern that there is not enough information for the reader regarding the use of size classes and also the use of the Rau, 1996 model. We now include further information below and this information is included in the supplementary information for the manuscript:

Size class calculations
The size classes of phytoplankton were calculated using seven diagnostic pigments which are used as biomarkers of specific taxa as calculated from the HPLC data (see methods). The taxa can be used to estimate the proportion of micro, nano and pico-phytoplankton. This is calculated using the following formulae:
$wDP=1.4(fucoxanthin) +1.41(peridinin) + 0.60(alloxanthin) + (0.35('19 –BF) + 1.27(19'-HF) + 0.86(zeaxanthin) +1.01(Chl b + divinyl - Chl b)$
$f_{micro} = (1.41(fucoxanthin) 1.41(peridinin)/wDP$
$f_{nano} = (0.60(alloxanthin) 0.35(19' - BF) 1.27(19' - HF)/wDP$

$f_{pico} = (0.86(zeaxanthin) 1.01(Chl b) divinyl - Chl b)/wDP$

The coefficients represent the average ratio between chla and the concentration of each diagnostic pigment, which are broadly related to taxa. This method contains caveats, which include:
- pigments are shared across taxa
- cells adjust their pigments ratios in response to light/nutrient stress
- this proxy was derived for a global study to estimate phytoplankton groups from satellites, therefore, the shifts in size structure as you go from the gyres (*Prochlorococcus* dominated) to an upwelling system (diatom dominated) are nicely captured but the high latitudes are misrepresented.

In this dataset we transition from gyre-like to mesotrophic conditions, which we believe should be accounted for relatively accurately with this method. Bricaud et al., 2004 also found a good correspondence to the optical properties of phytoplankton, which can be viewed as an independent proxy of cell size.

Rau et al., 1996 model

On initial experiments for this work, it was found that $[CO_{2(aq)}]$ alone was not a suitable determinant of the $\delta^{13}C$ of POC in surface waters across the SSTC, therefore the importance of other factors needed to be examined. The Rau model is used as the intracellular carbon concentration is dependent on $[CO_{2(aq)}]$, cell radius, cell growth rate, cell membrane permeability to $[CO_{2(aq)}]$ and temperature. This therefore allows the importance of these variables to be tested.

Here we include the baseline values within the model:

| Parameter | Value or calculation | Units |
|---|---|---|
| specific growth rate ($\mu$) | 1.1 | d-1 |
| instantaneous cell doubling time ($\mu_i$) | $\mu$/24/60/60 | d-1 |
| Enzymatic isotope fractionation associated with intracellular fixation ($\varepsilon f$) | 25 | ‰ |
| diffusive isotope fractionation of CO2aq in seawater ($\varepsilon d$) | 0.7 | ‰ |
| Cell wall permeability to $CO_2$ (P) | 1e-4 | m s$^{-1}$ |
| Surface area equivalent cell radius (r) | 10 | $\mu$m |
| Cell volume (V) | $(4\pi r^3)/3$ | $\mu$m$^3$ |
| Carbon content per cell ($\gamma c$) | 0.00000000000003154*V^0.758 | mol C |
| $CO_2$ uptake rate per cell ($Q_s$) | $(\gamma c*\mu_i)/(4*(pi*(r^2)))$ | mol C m$^{-2}$ s$^{-1}$ |
| Temperature –sensitive diffusivity of $CO_{2aq}$ in seawater ($D_T$) | 0.000005019*exp(-(19510/(8.3143*(temp+273.15)))) | m$^2$ s$^{-1}$ |
| $\delta^{13}C$ of $CO_2$ ($\delta^{13}C_{CO2}$) | 1.3+23.644-9701.5/(temp+273.15) | ‰ |
| $\delta^{13}C$ of particulate organic carbon ($\delta^{13}C$-POC) | $\delta^{13}C_{CO2}$-$\varepsilon f$+($\varepsilon f$-$\varepsilon d$)*(Qs*1e18) /(CO2*1000)*((r/1000000)/D$_T$+1/P) | ‰ |
| Uptake fractionation ($\varepsilon_p$) | $\delta^{13}C_{CO2}$ - $\delta^{13}C_{POC}$ | ‰ |

And a link to the MATLAB code for the model:
https://github.com/mvdh7/miscellanea/blob/master/g40s_isotopes/rau1996.m

**(1)      Page 6, Line 15 (and also page 7, line7): It is not clearly stated how model estimates based on 'temperature alone' are obtained, except for a reference to Rau et al. 1989. Is this the same as the original Farquhar model as described by François et al. ? If so, that model does not consider cell size .. but you mention a constant cell size of 10 µm was used. Please clarify.**

(2)      Using temperature alone signifies that all other baseline numbers within the model construct from Rau et al., 1996 have been used apart from the variability in temperature across the SSTC (see above table). The temperature is used to reconstruct CO2, which is used to predict variability in d13CPOC. All other variables within the model construct (see previous comment), are used as constants from the Rau model, therefore a constant cell size and growth rate.

*(3)      "To investigate the spatial variability across the SSTC, $[CO_{2(aq)}]$, and $\delta^{13}C_{CO2}$ were plotted against longitude and compared to model estimates (Rau et al., 1996, supplementary information), where we used the model constants for cell size (10 µm) and reconstructing $[CO_{2(aq)}]$ from temperature variability across the transect (Figure 3a and b)."*

**(1)      Page 7, Line 1 and Figure 5: authors state that cell radii were smaller in the subtropical waters compared to the SASW. From Figure 5 this is hardly visible.**

(2)      We thank the reviewer for highlighting this point, we agree that the colour plot makes it challenging to identify where the change in cell radii matches with the change in water mass. We attach a suggested amended figure with isotherms overlain for temperatures of 14 and 18 °C (Figure AC1). The smallest cell radii are within the core of the Agulhas and Brazil currents (where cell radii <8 µm and temperature >18 °C). The largest cell radii at the surface (excluding the Rio Plata) are within the SASW. Following on from a further comment below, we have also highlighted where stations were relative to the SSTC (north or south).

[Figure]

Figure AC1: Average cell radius across the SSTC (white contours show cell radius of 8 and 10µm). Black contours show temperatures of 14 and 18 °C. Green triangles mark the subtropical front with annotated regions north and south.

There is a significant moderate negative correlation between average cell radius and salinity (Figure AC2, Pearson's product moment correlation: r=-0.56, t = -8.69, df = 165, p-value = 3.36e-15). The lowest salinities from the Rio Plata have been removed from this (<33).

[Figure]

Figure AC2: Correlation between average cell radius and salinity, with temperature as colour.

**(1)        Page 8 Line 6: these trends 'contrast' the global observed variability. . . They contrast in what sense?**

(2)        This is an important point and we recognise that our wording needs to be clearer within the revised manuscript. This comment within the manuscript is describing the global trends: when CO2aq is high, Ep is high (i.e. in the Southern Ocean) and in areas where $[CO_{2(aq)}]$ is low such as the subtropical gyres, Ep is lower (see Figure 8). In our dataset the lowest $\varepsilon_p$ is where the $[CO_{2(aq)}]$ is the highest (and cell size is larger), see red/pink points in Figure 8a and d.

Suggested new wording:

(3)        *"Our data contrast the global observed variability (of high $\varepsilon_p$ in high $[CO_{2(aq)}]$ regions such as the Southern Ocean) but are comparable to results from previous work in frontal regions where higher $\varepsilon_p$ has been observed in lower $[CO_{2(aq)}]$ subtropical water masses (Bentaleb et al., 1998, Francois et al., 1993)."*

**(1)      Page 8, line 21: the sentence 'A higher growth rate increases the expression of a high εp on smaller phytoplankton' is unclear. Please reformulate.**

(2)      In Rau's 1996 model – when growth rate is higher, the effect of a variable surface area to volume ratio (or cell size) is expressed more on $\delta^{13}C_{POC}$. A higher growth rate increases the expression of a low $\varepsilon_p$ on larger phytoplankton compared to lower growth rates (Fry and Wainwright, 1991).

Wording changed to:

*(3) "A higher growth rate, such as in spring/summer blooms increases the range in $\varepsilon_p$ expressed across cell sizes. For instance in fast growing blooms, larger cell sizes may have higher relative $\delta^{13}C_{POC}$ and lower $\varepsilon_p$ than smaller cell sizes, compared to in low growth periods (e.g. Fry and Wainwright, 1991)."*

**(1)      Page 8 lines 20 to 27: the whole of the discussion here is highly hypothetical, and only yields a statement that waters north of the SSTC have 'the potential to elevate growth rates'. Later in the discussion it seems the 'potential for' has become a solid fact (e.g. line 14 and lines 19-20 at page 9).. Also it is likely that this frontal area is influenced by N-nutrient rich AAIW and SAMW waters. Have the authors considered this ?**
**Page 9, line20: Why would decreased light limitation lead to higher growth rates? Higher biomass and higher primary production, yes, but why higher growth rates?**

(2)      The nutrient rich SAMW (500m) and AAIW (750m) waters are deeper in the water column here, but the SASW originates from the surface waters of the polar frontal zone (ultimately sourced from the UCDW) and the northwards flowing waters which have high N in comparison to the subtropical waters (Tuerena et al., 2015). The SSTC creates an environment where there is the convergence of N-limited subtropical waters and Fe-limited subantarctic waters (Browning et al., 2014). This region therefore has the potential to alleviate nutrient stress. The convergence of water masses at the SSTC can also lead to strong and swift stratification and alleviation of light limitation, which would lead to higher growth rates (Llido et al., 2005). A study of the SSTC south of New Zealand found growth rates more than double the rates within the sub Antarctic and subtropical water masses (Delizo et al., 2007). We suggest that over the broader region, growth rates here will be higher across the SSTC than in the South Atlantic gyre or in the Southern Ocean.

**(1)      Page 9, Lines 16-20: increase cell size reduces the expression of a high εp as shown by the higher δ13CPOC and lower εp ..). It seems to me the data points rather fit the general trend of δ13C and ep, and highlighted offset mentioned, appears weak.**

(2)      The data points fit the trend to a lesser degree than expected for $[CO_{2(aq)}]$. Note the red-pink points in 8a have a higher than predicted $[CO_{2(aq)}]$ at the given latitude (40-50°S) and a lower than predicted $\delta^{13}C_{CO2}$. It would be intuitive therefore to predict that the $\delta^{13}C_{POC}$ produced would be lower than the average trend, but is in fact higher. $\varepsilon_p$ is also lower than predicted with all of the cell radii $>10\mu m$.

**(1)      Though this is not the subject of this paper, it is interesting to see this decrease of δ13C DIC also in cold North Atlantic waters. What is the explanation for this phenomenon?**

The lower $\delta^{13}C$ in the North Atlantic and Southern Ocean is related to circulation and the relative extent of photosynthesis and respiration of nutrients and carbon within surface waters. In the low latitude ocean, nutrients and DIC are much lower in the surface ocean from downwelling and the uptake of nutrients and regeneration at depth, therefore $\delta^{13}C_{DIC}$ is higher in the lower latitudes compared to the higher latitudes. The concentrations of DIC and nutrients are higher in the Southern Ocean compared the North Atlantic (more upwelling), therefore the $\delta^{13}C_{POC}$ is lower relative to the North Atlantic.

**(1)      Page 10, line 5: ".. predicting increases in εp and decreases in δ13CPOC " . Figure 9 rather shows increasing temperature would result in decreased εp and increased δ13CPOC ..**
On page 12, line 5.

Although an increase in temperature in the figure shows an increase in $\delta^{13}C_{POC}$ and a decrease in $\varepsilon_p$, this will have very little effect compared to the predicted changes in carbon availability and cell size. To give an example:

A 2°C change in SST from 14 to 16°C would increase $\delta^{13}C_{POC}$ from -23.9‰ to -23.3‰. That is the predicted change over ~200yrs (IPCC). Over this time period atmospheric $CO_2$ would increase from

pre-industrial to 500ppm which would decrease $\delta^{13}C_{POC}$ to -26‰ (at 14°C) and -25.5‰ (at 16°C). Decreasing cell radius from 10μm to 8μm would decrease $\delta^{13}C_{POC}$ further to -27‰ (14°C) and -26.5‰ (16°C).

Therefore a 2°C increase in SST with the expected rise in atmospheric $CO_2$ would decrease $\delta^{13}C_{POC}$ from -23.9‰ to -25.5‰ and would decrease further if the average cell size decreased.

Please see the following paragraph towards the end of the discussion:

*"Seawater warming, which is expected to accompany future increases in $[CO_{2(aq)}]$, independently modulates the marine carbonate system (Humphreys, 2017) and the fractionation model of Rau et al. (1996). In this case, simultaneous warming would oppose the increase in $\varepsilon_p$ (and therefore decrease in $\delta^{13}C_{POC}$) driven by increasing $[CO_{2(aq)}]$, as shown by the negative line gradients in Figure 9a (and positive gradients in Figure 9b). However, this is expected to have a relatively small impact overall, as the following back-of-the-envelope calculation illustrates. Given an equilibrium climate sensitivity (i.e. the equilibrium warming of Earth's near-surface resulting from a doubling of atmospheric $pCO_2$) of 1.5 to 4.5 °C (Stocker et al., 2013), an increase in $pCO_2$ from 400 to 500 ppm would drive from 0.5 to 1.5 °C of global mean warming. For 10 μm cells, the $pCO_2$ change alone would increase $\varepsilon_p$ by ~1.8 ‰, while this warming alone would decrease $\varepsilon_p$ by only 0.1 to 0.4 ‰, according to the model of Rau et al. (1996)."*

**Page 12: the authors conclude to the significance of their findings for future studies of δ13C in food web studies. They could add that this also extends to future studies about the fate of plankton organic matter in the deep ocean. In that aspect an useful paper that can be cited is the one by Cavagna et al., BG 10, 2013 "Water column distribution and carbon isotopic signal of cholesterol, brassicasterol and POC in the Atlantic sector of the S.O."**

We thank the reviewer for this useful addition, please see amended text:

*"Therefore the factors which contribute to variability at the base of the food web need to be understood well in order to accurately understand marine food web dynamics (Peterson and Fry, 1987). These findings could also have implications to the distribution of $\delta^{13}C_{POC}$ in the deep ocean through organic matter sinking and burial (e.g. Cavagna et al., 2013)."*

**Minor things:**
**(1)      Page 4 line 6: 50 ml of 100% HgCl₂ were added; I guess you mean 50 μl ..?**
(2)      Correct, this has been amended
*(3)      "and 50 μL of 100% HgCl₂ added"*

**(1)      Page 10, line 11: the wording 'physiological status' is rather vague.. can you specify more ?**
Changed to:
*(3)      "including the physiological dependencies of phytoplankton on light and nutrients and their ecological diversity"*

**(1)      Figures 4 and 5: mark the waters located north and south of the SSTC**
(2)      Orange triangles have been added to the figures to mark the SSTC.

[Figure]

**(1)** **Figure 7: the full red line is not specified**

(2) See edited figure below:

[Figure]

Response to Reviewer 1b

(1) 1/ A further comment concerning the following reply of the authors (page C7): "Although an increase in temperature in the Figure shows an increase in δ13CPOC and a decrease in ep, this will have very little effect compared to the predicted changes in carbon availability and cell size. "
I suggest authors make this future change (decrease) in d13CPOC more visible to the reader by marking it in Figure 9b. For example they could mark the jump from the 400 ppm to the 500ppm level with increasing temperature by an arrow.
(2) We have added an arrow to Figure 9b to include the projected change from 100ppm $CO_2$ increase and a 2°C temperature increase (cell radius 10μm).

(1) 2/ In their reply on the question about the latitudinal distribution of d13C-DIC, the authors don't really clarify the issue, I believe. Of course Southern Ocean d13C-DIC is very low because of upwelling of deep ocean waters depleted in 13C-DIC there, a phenomenon not present in the North Atlantic. So I feel the question about which process really imposes lower d13C-DIC in the North Atlantic is not satisfactorily resolved by their reply. Admitedly this is not the subject of their paper.
(2) We include a Figure to show the relationship between d13C-CO2 and CO2aq globally, using the data from Figure 8. The d13C falls in line with expected values for the given CO2aq of the North Atlantic (~-9‰, red points in Figure1b). The Southern Ocean values are lower than the North Atlantic due to upwelling (~-10-11‰), we will expand the axes in Figure 8b to make it more apparent.

[Figure]

Figure 1, relationship between $\delta^{13}C_{CO2}$ and $[CO_{2aq}]$. (a) map of data points, (b) $\delta^{13}C_{CO2}$ and $[CO_{2aq}]$ with latitude as a z variable, (c) $\delta^{13}C_{CO2}$ and $[CO_{2aq}]$ with longitude as a z variable.

Response to reviewer 2.
We thank the reviewer for their time in completing this review, we believe that their input will help greatly improve the manuscript. Here we include responses to all of the comments:
    (1) Reviewer's comment
    (2) Author's comment
    (3) Suggested change to manuscript

**(1) This is an interesting paper looking at the variability in carbon isotope (and fractionation) of particulate organic matter (with CO2aq) in relation to phytoplankton cell size. The authors sampled subantarctic and subtropical regimes with contrasting environments and community structures to investigate mechanisms for isotopic fractionation in d13CPOC resulting from carbon uptake and biological production in the upper ocean. The authors suggest that cell size is an important factor. Using estimates of cell size (via HPLC analyses) and calculated CO2aq, the authors suggest that smaller cells will respond less to increased CO2aq than the larger cells south of the SSTC and the wider Southern Ocean.**

**Query: when looking at investigating future epsilon-p did the authors consider the combined effect of increased CO2 and increased temperature in the two environments?**

(2) We refer to our response to reviewer 1, which describes the expected changes to temperature as well as $CO_2$ increases (the temperature increases would have a much lesser effect than $CO_2$ increases and a decrease in cell size).

'Although an increase in temperature in the figure shows an increase in $\delta^{13}C_{POC}$ and a decrease in ep, this will have very little effect compared to the predicted changes in carbon availability and cell size. To give an example:

A 2°C change in SST from 14 to 16°C would increase $\delta^{13}C_{POC}$ from -23.9‰ to - 23.3‰, which is the predicted change over ~200yrs (IPCC). Over this time period atmospheric CO2 would increase from pre-industrial to 500ppm which would decrease $\delta^{13}C_{POC}$ to -26‰ (at 14°C) and -25.5‰ (at 16°C). Decreasing cell radius from 10um to 8um would decrease $\delta^{13}C_{POC}$ further to -27‰ (14°C) and -26.5‰ (16°C).

Therefore a 2°C increase in SST with the expected rise in atmospheric $CO_2$ would decrease $\delta^{13}C_{POC}$ from -23.9‰ to -25.5‰ and would decrease further if the average cell size decreased.'

This has been discussed in the manuscript:
*"Seawater warming, which is expected to accompany future increases in [CO$_{2(aq)}$], independently modulates the marine carbonate system (Humphreys, 2017) and the fractionation model of Rau et al. (1996). In this case, simultaneous warming would oppose the increase in ε$_p$ (and therefore decrease in δ$^{13}$C$_{POC}$) driven by increasing [CO$_{2(aq)}$], as shown by the negative line gradients in Figure 9a (and positive gradients in Figure 9b). However, this is expected to have a relatively small impact overall, as the following back-of-the-envelope calculation illustrates. Given an equilibrium climate sensitivity (i.e. the equilibrium warming of Earth's near-surface resulting from a doubling of atmospheric pCO$_2$) of 1.5 to 4.5 °C (Stocker et al., 2013), an increase in pCO$_2$ from 400 to 500 ppm would drive from 0.5 to 1.5 °C of global mean warming. For 10 μm cells, the pCO$_2$ change alone would increase ε$_p$ by ~1.8 ‰, while this warming alone would decrease ε$_p$ by only 0.1 to 0.4 ‰, according to the model of Rau et al. (1996)."*

But we have further included an arrow in Figure 9b to demonstrate the combined effects of increased temperature and [CO$_{2(aq)}$].

**(1) General point about Figures, it is very hard to deduce where measurements were taken in the profiles and also which interpolations were used to create the profiles.**

(2) We have edited Figures 1 and 4 to have more visible points in the profiles (larger point size). The interpolation for these figures has been made using ODV and the weighted average gridding (x, y spacing determined by profile spacing). Information about this has now been included into the captions.

**(1) Initial thoughts while starting to read the manuscripts were: 'but what about species composition'? This really only gets dealt with in the discussion. It would be good to see this upfront, including a small discussion about cell size on its own (so possibly discussing culture studies) actually supports what the authors conclude.**

(2) We have edited the introduction to include more information about cell size and the implications to carbon isotope studies:
End of first paragraph:
*"Alterations to phytoplankton diversity and/or productivity will likely have knock-on effects on marine food web dynamics. Investigating such changes in the remote marine environments requires tracers that can pinpoint shifts in dietary sources. The $\delta^{13}C$ of organic carbon in marine plants and animals can provide information on carbon sources to the base of the food web (Peterson and Fry, 1987, Post, 2002). Improved understanding of $^{13}C$ systematics will lead to reliable use of this proxy in future predictions."*
Fourth paragraph:

*"Phytoplankton growth rate, cell size and cell geometry are also important controls on $\delta^{13}C_{POC}$ in surface waters (Bidigare et al., 1997; Francois et al., 1993; Popp et al., 1998; Laws et al., 1995; Villinski et al., 2000). These ecophysiological factors decouple the observed relationship between $\delta^{13}C_{POC}$ and $[CO_{2(aq)}]$, limiting the reliability of $\delta^{13}C_{POC}$ as a palaeoproxy. This is particularly true in areas where $[CO_{2(aq)}]$ is lower or less variable, as other factors have been found to be more important for determining the degree of isotopic fractionation (Henley et al., 2012; Lourey et al., 2004; Popp et al., 1998). In field studies, smaller sized phytoplankton have been measured with lower $\delta^{13}C_{POC,}$ compared to larger cells such as diatoms, particularly in fast growing blooms (Hansman and Sessions, 2015, Rau et al., 1990). These findings indicate that the factors determining $\delta^{13}C_{POC}$ may vary as you transition contrasting marine environments."*

**(1) Introduction:**
**Second sentence: missing a bit; anthropogenic CO2 input to the atmosphere causes enhanced greenhouse gasses, which causes the oceans to warm up. It is not a direct effect.**

(2) Sentence changed to:

(3) *"Anthropogenic carbon inputs and the increase of greenhouse gases in the atmosphere are causing ocean warming (Cheng et al., 2019), changes to upper ocean stratification (Bopp et al., 2001; Capotondi et al., 2012) and altered distributions of nutrients and carbon (Khatiwala et al., 2013; Quay et al., 2003; Gruber et al., 2019)."*

**(1) Methods: A bit strange to see details of where the inorganic carbon isotopes where analysed, but none of the other analyses.**

(2) We agree with the reviewer and have removed 'University of Cambridge' from the manuscript. Sentence now reads:
(3) *"Samples were measured using a Thermo MAT253 stable isotope mass spectrometer."*

**(1) Results: 3.1 first para. In reference to Figure 1, what does MC stand for?**

(2) Sentence changed to:
(3) *"The three subtropical water masses (Agulhas Current (AC), South Atlantic Central Water (SACW) and Brazil Current (BC)) can be readily identified with warmer temperatures and higher salinities, the influence of the Malvinas Current (MC) separates the core of the SACW and BC (Figure 1)."*

**(1) Figure 1 does not show a correlation between various variables, just cross sections.**

(2) Sentence has been edited to read:
(3) *"Across the zonal transect, higher $\delta^{13}C_{CO2}$ is associated with lower [CO2(aq)] and warmer temperatures of the subtropical water masses (Figure 1)."*

**(1) 3.2 Para 3 'There is no significant correlation between d13CPOC and CO2aq or d13CCO2 (Fig 2)' where? Subtropical samples?**

(2) Sentence has been edited to read:
(3) *"There are no significant correlations between $\delta^{13}C_{POC}$ and [CO_{2(aq)}] or $\delta^{13}C_{CO2}$ in the subtropical or subantarctic water masses (Figure 2, p>0.05)."*

**(1) Para 4 Statement: Picoplankton were dominant in the subtropical environments. NO. This figure suggests that fmicro and fnano are dominant in all environments.**

(2) We thank the reviewer for highlighting this error in our wording and have changed the sentence accordingly:
(3) *"Picophytoplankton were more abundant in the subtropical environments in comparison to the SASW, contributing between 30-40% of the pigment biomass at the core of these water masses (Figure 4)."*

**(1) The authors claim there is a significant positive correlation between average community cell radius and d13CPOC, with n=30. There are 47 data points in Figure 6a; in Figure 6b 4 are attributed to being coastal sites. What happened to the missing 13 data points?**
There is less data in Figure 6b as we did not have corresponding cell size data for all of the $\delta^{13}C_{POC}$ data points, to inform the reader, this information has been added to the figure caption.

**(1) Page 7: with to first sentence and reference to Figure 5: what is the average error and is the suggested difference supported by statistics?**

In general there is no significant difference between the two water masses when you take the definitions of >14 and <14C for subtropical and subantarctic (south and north of the SSTC), as there is the convergence and mixing of water masses in this region. The large errors associated with the average cell radii can arise from the variation at the DCM of the subtropical water masses (larger size cells) and the variability from the mixing of water masses and thus different nutrient requirements. If we use only the cores of each of the surface water masses and discount the variability at the DCM, then there is a significant difference (Subtropical >20C 6.5 ±0.8, n17, Subantarctic<18C 10.4 ±2.3 n31 ). Because of this ambiguity, we change the wording accordingly:

(3) *"Estimated average cell radii were generally smaller at the core of the subtropical water masses compared to the SASW (Figure 5) (depth range <40 m, subtropical [>20 °C] 6.5 $\mu m$ ±0.8, n=17, subantarctic [<18 °C] 10.4 $\mu m$ ±2.3, n=31)."*

**(1) Discussion: add some references when discussing the used of stable isotopes of organic**

**matter as a primary means for examining food web structure and variability. Plus also to line 32-33 (nitrogen isotopes).**

This is a valuable comment we have added extra references to the text:

[revised manuscript text omitted]

**Supplementary information for Carbon uptake**

**Size class calculations**

The size classes of phytoplankton were calculated using seven diagnostic pigments which are used as biomarkers of specific taxa as calculated from the HPLC data (see methods). The taxa can be used to estimate the proportion of micro-, nano- and picophytoplankton. This is calculated using the following formulae:

$wDP = 1.4(\text{fucoxanthin}) + 1.41(\text{peridinin}) + 0.60(\text{alloxanthin}) + (0.35(19' - BF) + 1.27(19' - HF) + 0.86(\text{zeaxanthin}) + 1.01(\text{Chl } b + \text{divinyl} - \text{Chl } b)$

$f_{micro} = (1.41(\text{fucoxanthin})\ 1.41(\text{peridinin})/wDP$

$f_{nano} = (0.60(\text{alloxanthin})\ 0.35(19' - BF)\ 1.27(19' - HF)/wDP$

$f_{pico} = (0.86(\text{zeaxanthin})\ 1.01(\text{Chl } b)\ \text{divinyl} - \text{Chl } b)/wDP$

The coefficients, derived from multiple regression analysis of chla and the concentration of the most dominant diagnostic pigments, are broadly related to taxa. This method contains caveats, which include:

- pigments are shared across taxa
- cells adjust their pigments ratios in response to light/nutrient stress
- this proxy was derived for a global study to estimate phytoplankton groups from satellites, therefore, the shifts in size structure as you go from the gyres (*Prochlorococcus* dominated) to an upwelling system (diatom dominated) are nicely captured but the high latitudes may be misrepresented.

In this dataset we transition from gyre-like to mesotrophic conditions, which we believe should be accounted for relatively accurately with this method. Bricaud et al., 2004 also found a good correspondence to the optical properties of phytoplankton, which can be viewed as an independent proxy of cell size.

Rau et al., 1996 model

On initial experiments for this work, it was found that $[CO_{2(aq)}]$ alone was not a suitable determinant of the $\delta^{13}C$ of POC in surface waters across the SSTC, therefore the importance of other factors needed to be examined. The Rau model is used as the intracellular carbon concentration is dependent on $[CO_{2(aq)}]$, cell radius, cell growth rate, cell membrane permeability to $[CO_{2(aq)}]$ and temperature. This therefore allows the importance of these variables to be tested.

Here we include the baseline values within the model:

| Parameter | Value or calculation | Units |
|---|---|---|
| specific growth rate ($\mu$) | 1.1 | d-1 |
| instantaneous cell doubling time ($\mu_i$) | $\mu$/24/60/60 | d-1 |
| Enzymatic isotope fractionation associated with intracellular fixation ($\varepsilon f$) | 25 | ‰ |
| diffusive isotope fractionation of CO2aq in seawater ($\varepsilon d$) | 0.7 | ‰ |
| Cell wall permeability to $CO_2$ (P) | 1e-4 | m s$^{-1}$ |
| Surface area equivalent cell radius (r) | 10 | $\mu$m |
| Cell volume (V) | $(4\pi r^3)/3$ | $\mu$m$^3$ |
| Carbon content per cell ($\gamma c$) | 0.00000000000003154*V^0.758 | mol C |
| $CO_2$ uptake rate per cell ($Q_s$) | $(\gamma c*\mu_i)/(4*(pi*(r^2)))$ | mol C m$^{-2}$ s$^{-1}$ |
| Temperature –sensitive diffusivity of $CO_{2aq}$ in seawater ($D_T$) | 0.000005019*exp(-(19510/(8.3143*(temp+273.15)))) | m$^2$ s$^{-1}$ |
| $\delta^{13}C$ of $CO_2$ ($\delta^{13}C_{CO2}$) | 1.3+23.644-9701.5/(temp+273.15) | ‰ |
| $\delta^{13}C$ of particulate organic carbon ($\delta^{13}C$-POC) | $\delta^{13}C_{CO2}$-$\varepsilon f$+($\varepsilon f$-$\varepsilon d$)*(Qs*1e18) /(CO$_2$*1000)*((r/1000000)/D$_T$+1/P) | ‰ |
| Uptake fractionation ($\varepsilon_p$) | $\delta^{13}C_{CO2}$ - $\delta^{13}C_{POC}$ | ‰ |

And a link to the MATLAB code for the model:
https://github.com/mvdh7/miscellanea/blob/master/g40s_isotopes/rau1996.m